# The electron–proton bottleneck of photosynthetic oxygen evolution

Paul Greife[1,4], Matthias Schönborn[1,4], Matteo Capone[2,3,4], Ricardo Assunção[1], Daniele Narzi[3], Leonardo Guidoni[3✉] & Holger Dau[1✉]

Photosynthesis fuels life on Earth by storing solar energy in chemical form. Today's oxygen-rich atmosphere has resulted from the splitting of water at the protein-bound manganese cluster of photosystem II during photosynthesis. Formation of molecular oxygen starts from a state with four accumulated electron holes, the $S_4$ state—which was postulated half a century ago[1] and remains largely uncharacterized. Here we resolve this key stage of photosynthetic $O_2$ formation and its crucial mechanistic role. We tracked 230,000 excitation cycles of dark-adapted photosystems with microsecond infrared spectroscopy. Combining these results with computational chemistry reveals that a crucial proton vacancy is initally created through gated sidechain deprotonation. Subsequently, a reactive oxygen radical is formed in a single-electron, multi-proton transfer event. This is the slowest step in photosynthetic $O_2$ formation, with a moderate energetic barrier and marked entropic slowdown. We identify the $S_4$ state as the oxygen-radical state; its formation is followed by fast O–O bonding and $O_2$ release. In conjunction with previous breakthroughs in experimental and computational investigations, a compelling atomistic picture of photosynthetic $O_2$ formation emerges. Our results provide insights into a biological process that is likely to have occurred unchanged for the past three billion years, which we expect to support the knowledge-based design of artificial water-splitting systems.

In all plants, algae and cyanobacteria, sunlight drives the splitting of water molecules into energized electrons and protons, both of which are needed for the reduction of $CO_2$ and eventually carbohydrate formation[2]. Molecular oxygen ($O_2$) is formed during this process, which transformed the Earth's atmosphere during the 'great oxygenation event'[3], which began about 2.4 billion years ago. Light-driven water oxidation occurs at the oxygen-evolving complex, a $Mn_4CaO_5$ cluster bound to the proteins of photosystem II[2,4] (PSII). The relationship between electron and proton transfer in the bottleneck steps of $O_2$ formation has remained incompletely understood. We address this key step here using time-resolved Fourier transform infrared (FTIR) experiments (Fig. 1).

## Time-resolved tracking of $O_2$ transition

To perform time-resolved infrared spectroscopy on PSII, we developed an FTIR step-scan experiment with automated exchange of dark-adapted PSII particles (Methods), thereby expanding previous experiments at individual wavenumbers[5–7] towards detection of complete fingerprint spectra. The sample exchange system was refilled about every 60 h using PSII membrane particles with about 1.5 g of chlorophyll prepared from 40 kg of fresh spinach leaves for day and night data collection over a period of 7 months. We initiated the transitions between semi-stable S states by 10 visible light (532 nm) nanosecond laser flashes applied to the dark-adapted photosystems (Extended Data Fig. 1). Using a specific deconvolution approach based on Kok's standard model[1] (Fig. 1a), we obtained time-dependent S-state difference spectra for each of the individual transitions between the four semi-stable reaction-cycle intermediates $S_1$, $S_2$, $S_3$ and $S_0$ (for selected time courses see Extended Data Fig. 2).

We focus on the oxygen-evolution transition, $S_3 \rightarrow S_4 \rightarrow S_0 + O_2$, predominantly induced by the third laser flash, for which time courses at selected wavenumbers are shown in Fig. 2a (time-resolved spectra are shown in Extended Data Fig. 3). Multiexponential simulations of the time courses provided 5 time constants describing acceptor- and donor-side PSII processes, including the expected time constants of 340 μs and 2.5 ms. The 2.5-ms time constant ($t_{O_2}$) corresponds to the reciprocal rate constant of the rate-determining step in O–O bond formation and $O_2$ release[8,9]. The 340 μs time constant ($t_{H^+}$) corresponds to an obligatory step of proton removal from the oxygen-evolving complex of PSII, as shown recently by time-resolved detection of X-ray absorption, UV-visible spectroscopy, recombination fluorescence and photothermal signals[10–12], resulting in a specific $Mn(IV)_4 Tyr_Z^{ox\bullet}$ metalloradical intermediate that was also trapped in low-temperature magnetic resonance experiments[13,14]. 'Obligatory' here signifies that the O–O bond formation chemistry can proceed only after proton removal is complete, as verified by the delayed onset of signals that trace manganese oxidation states or, generally, the $O_2$ formation chemistry[8,10,12,15,16],

[1]Department of Physics, Freie Universität, Berlin, Germany. [2]Department of Information Engineering, Computer Science and Mathematics, University of L'Aquila, L'Aquila, Italy. [3]Department of Physical and Chemical Sciences, University of L'Aquila, L'Aquila, Italy. [4]These authors contributed equally: Paul Greife, Matthias Schönborn, Matteo Capone. ✉e-mail: leonardo.guidoni@univaq.it; holger.dau@fu-berlin.de

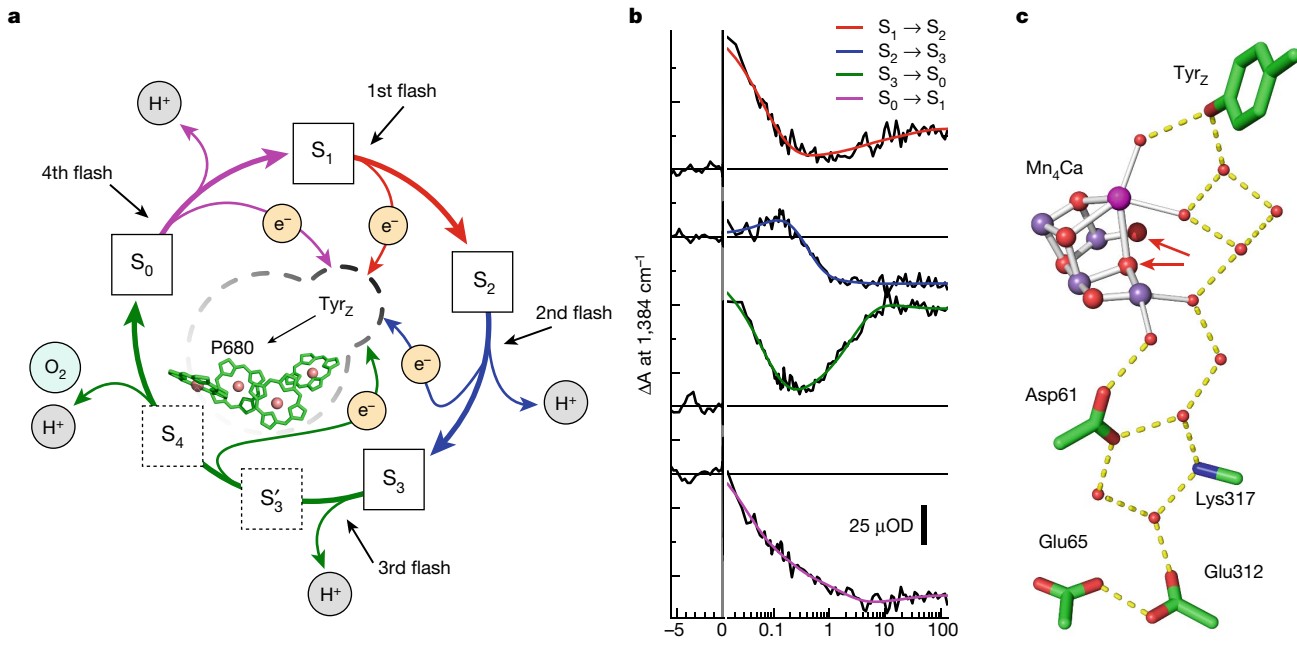

**Fig. 1 | Reaction cycle of photosynthetic oxygen evolution. a**, Model of the S-state cycle with sequential electron and proton removal from the oxygen-evolving site[10,11,50]. Starting in the dark-stable $S_1$ state, each laser flash initiates oxidation of the primary chlorophyll donor ($P680^+$ formation) followed by electron transfer from a tyrosine sidechain ($Tyr_Z$ oxidation) and—in three of the four S-state transitions—manganese oxidation, until four electron holes (oxidizing equivalents) are accumulated by the $Mn_4Ca$-oxo cluster in its $S_4$ state. **b**, Example of tracing S-state transitions using IR absorption changes after excitation with visible-wavelength laser flashes (at zero on the time axis). The absorption changes ($\Delta A$) are provided in optical density (OD) units. The IR transients at 1,384 cm$^{-1}$ reflect symmetric stretching vibrations of carboxylate

protein sidechains that sense changes in the oxidation state of manganese in the microsecond and millisecond time domain (coloured lines are simulations with time constants provided in Supplementary Table 2). Note that the scale on the $x$ axis is linear below $t = 0$ and logarithmic above $t = 0$. **c**, The $Mn_4Ca$ cluster (Mn, violet; Ca, pink) in the $S_3$ state with six bridging oxygens, the redox-active tyrosine ($Tyr_Z$), and further selected protein sidechains as well as water molecules (red spheres), based on crystal structures[25]. Assignment to polypeptide chains, numbering of the atoms of $Mn_4Ca$-oxo and water molecules and hydrogen-bond distances are indicated in Supplementary Fig. 1. The two oxygens atoms that form the O–O bond in the oxygen-evolving $S_3 \rightarrow S_0$ transition are indicated by red arrows.

which is also visible in the top time course of Fig. 2a. For systematic analysis of the 2D time–wavenumber data array obtained by the FTIR step-scan experiment, we exploited that the requirement for wavenumber independence of the time constants of proton removal ($t_{H^+} = 340$ µs) and the electron transfer associated with O$_2$ formation ($t_{O_2} = 2.5$ ms), because they always reflect the same reaction (the same rate constant). The time constants can thus serve as a kinetic tag of the reaction in the time-resolved spectroscopic data. By simultaneous simulation of the time courses at 2,582 wavenumbers (1,800 cm$^{-1}$ to 1,200 cm$^1$) using the same set of time constants at each wavenumber, we obtained the amplitude spectra shown in Fig. 2b–d, which are denoted as decay-associated spectra (DAS).

## Pivotal sidechain deprotonation

Conventional steady-state S-state difference spectra collected hundreds of milliseconds after the laser flash could reflect the changes directly coupled to the stable light-induced oxidation-state changes of manganese ions and sensed by the coordinated protein sidechains, as often assumed[17,18] (but see also ref. 19). Then, the steady-state difference spectrum should correspond to the DAS of the $t_{O_2}$ component (2.5 ms), as indeed visible in Fig. 2d, supporting assignment to the Mn- or Ca-binding carboxylate sidechains. In other spectral regions, however, we observed a different behaviour: in the $t_{O_2}$ amplitude spectrum, (1) positive peaks at 1,700–1,750 cm$^{-1}$ (Fig. 2b) and (2) negative peaks at 1,565–1,605 cm$^{-1}$ (Fig. 2c) are not matched by counterparts in the steady-state spectrum. These peaks are probably assignable to (1) vibrations of protonated carboxylate sidechains (C=O vibrations

of Asp- or Glu-associated carboxylic acid groups) and (2) deprotonated sidechains of carboxylates[20] (asymmetric O–C–O vibrations of Asp- or Glu-associated carboxylate ions). The mismatch between $t_{O_2}$ amplitudes and the near-zero steady-state spectrum is explainable by carboxylate sidechain deprotonation early in the $S_3 \rightarrow S_4 \rightarrow S_0$ transition, which is subsequently reversed by reprotonation in parallel with O$_2$ formation, as indicated by the inverted amplitude spectra of the $t_{H^+}$ and $t_{O_2}$ components (marked by red areas). On these grounds, and supported by further analyses including experiments in deuterated water (Supplementary Figs. 9 and 10), we conclude that carboxylate deprotonation coincides with the relocation of a proton towards the aqueous solvent ($t_{H^+}$) and carboxylate reprotonation paralleling the O$_2$ formation step ($t_{O_2}$).

The presence of the three well-resolved bands at 1,707, 1,723 and 1,744 cm$^{-1}$ (Fig. 2b and Extended Data Fig. 4) is attributed to a specific pair of carboxylate residues with fluctuating hydrogen-bond configurations: the Glu65–Glu312 pair in Fig. 1c, which may be described as a proton gate[21] or proton-loading site[22,23]. In analogy to the proton-loading site of cytochrome $c$ oxidase[24], fractional protonation state changes of further groups might also be involved. Next, on the basis of a recent crystallographic model of the oxygen-evolving complex in its $S_3$ state[25], we explore a plausible carboxylate assignment and track the O–O bond formation computationally, starting from the $Tyr_Z^{ox}$ $S_3$ state.

## Atomistic scenario by quantum chemistry

We investigated the reaction path leading to O–O bond formation using density functional theory (DFT)-based minimum energy path (MEP)

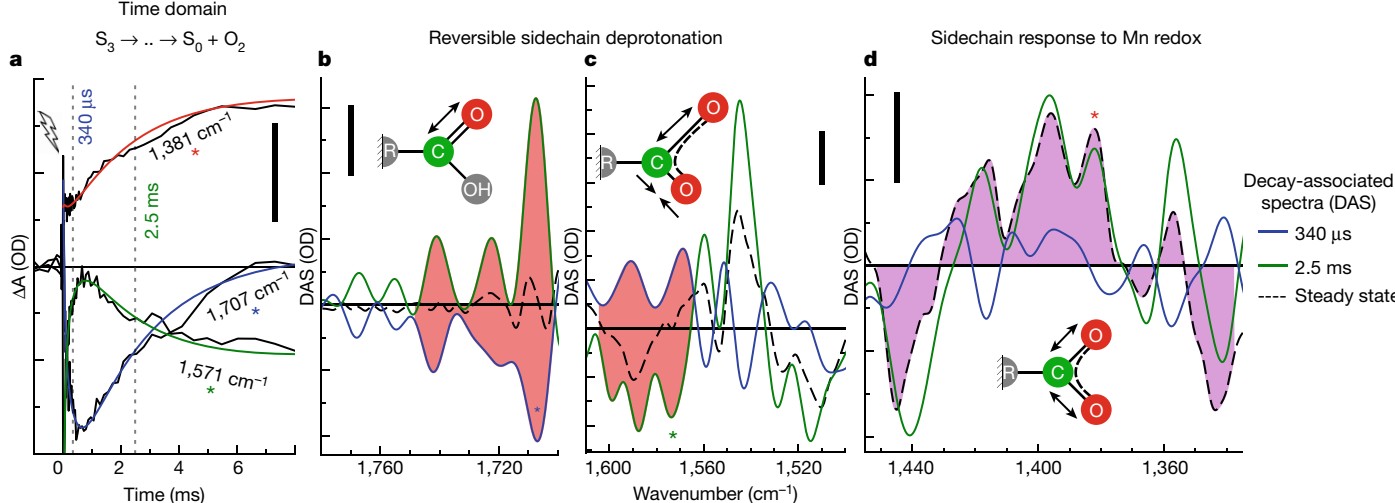

**Fig. 2 | Oxygen-evolution transition traced by FTIR. a**, IR time traces at selected wavenumbers, demonstrating the delayed onset of O–O bond formation (1,381 cm⁻¹) and reversible changes assignable to transient sidechain deprotonation (1,571 cm⁻¹ and 1,707 cm⁻¹). The corresponding wavenumbers in the spectra in **b**–**d** are marked with coloured asterisks. **b**–**d**, DAS corresponding to the proton release phase ($t_{H^+}$ = 340 μs, blue line)

and the oxygen-evolution phase ($t_{O_2}$ = 2.5 ms, green) as well as the steady-state difference spectrum of the $S_3 \rightarrow S_0 + O_2$ transition (dashed black line). Red areas **b**,**d** mark inverted 340 μs DAS and 2.5 ms DAS, indicating reversible behaviour; purple shaded areas in **c** highlight the similarity of the 2.5 ms DAS and the steady-state spectrum, in line with the assignment to non-transient changes in Mn oxidation state. Scale bars, 50 μOD.

calculations (see Methods). The computational tracing of O–O bond formation starts with an oxidized redox-active tyrosine (Tyr$_Z$), and the Mn$_4$Ca cluster still in its $S_3$-state conformation[26] (Extended Data Fig. 6), which has previously been determined by serial free-electron laser crystallography[25,27]. Moreover, it is assumed that the transition from the $S_3$ to the $S_3'$ state has been completed (with time constant of 340 μs), resulting in a deprotonated state of a carboxylate group, which is the prerequisite for the subsequent reactions leading via the $S_4$ state to O–O bond formation and $O_2$ release.

The calculated MEP comprises three metastable states (i, ii and iii) and two transitions as shown in Fig. 3. In the first transition (i → ii), electron transfer (to Tyr$_Z^{ox}$) is coupled to proton transfer from a Mn-bound hydroxide (Mn1(IV)-O6H) to a specific water molecule (W2), resulting in an oxyl radical terminally bound to a Mn ion (Mn1(IV)-O6˙). The transfer of the Tyr$_Z^{ox}$ hole to O6 constitutes oxidation of a substrate-water oxygen by one electron. The thereby reached $S_4$ state cannot be considered a transition state, but rather a reaction intermediate which is stable (minimally) over the 10-ps trajectory of the ab initio molecular dynamics simulations[28]. Eventually, the final state (iii) is reached by oxo–oxyl coupling and peroxide formation. In contrast to previous investigations[26,29–31], our MEP calculations on the i → ii transition are not restricted to the local events at the Mn$_4$Ca-oxo cluster and its immediate ligand environment. They also include the electron transfer step (Supplementary Information)—that is, the simultaneous Tyr$_Z$ reduction and O6 oxidation, as well as the coupled shift of four protons from initial hydrogen-bond donor to hydrogen-bond acceptor. The nuclear rearrangements around Tyr$_Z$ and the Mn$_4$Ca-oxo cluster are not isolated events, but are interlinked via a cluster of hydrogen-bonded water molecules that is included in the quantum region of the MEP calculations.

The obvious mechanistic problem of the $S_3' \rightarrow S_4$ transition is that both (1) Mn(IV)-OH deprotonation without previous hole transfer to Tyr$_Z^{ox}$ and (2) hydroxyl radical formation without previous deprotonation would involve an energetically unfavourable intermediate state (Supplementary Information). Now our MEP calculations show how this mechanistic problem can be solved via the coupled (concerted) movement of one electron (to Tyr$_Z^{ox}$) and three protons (Fig. 3c,d and Supplementary Video). This proton movement can be viewed as a Grotthus-type proton transfer: the individual protons are shifted within a hydrogen bond by only about 1 Å, whereas overall the protonation is

moved over a distance of approximately 7 Å, from O6 to Asp61. Additional structural information about these and further steps of the $S_3 \rightarrow S_0$ transition are reported in Extended Data Fig. 8. We emphasize that the reaction coordinates presented here extend previous computational investigations[30–34] by including both the pivotal protonation and hydrogen-bond dynamics associated with Tyr$_Z$ reduction and oxyl-radical formation, thereby enabling a plausible connection to the experimentally detected entropy of activation (Supplementary Information).

The MEP calculation results in an energetic barrier of Mn(IV)-O˙ formation in the $S_3' \rightarrow S_4$ transition of only 7 kcal mol⁻¹ (300 meV). This figure agrees surprisingly well with the experimental value of 7.1 kcal mol⁻¹ (310 meV) that we determined for the crystallographically characterized cyanobacterial photosystems from *Thermosynechococcus elongatus* from the temperature dependence of the rate constant of $O_2$ formation (Arrhenius plot) by applying Eyring's transition-state theory (Extended Data Fig. 5). Assuming a purely enthalpic free energy of activation, these values would imply a time constant below 1 μs, whereas we detected a millisecond time constant. The experimental findings thus indicate a substantial entropic contribution to the free energy of activation of about 6.5 kcal mol⁻¹ (285 meV), which corresponds to a slowdown by a factor exceeding 10,000. The pronounced entropic slowdown is most plausibly explained by many approximately isoenergetic conformations of the hydrogen-bonded protein–water network at the active site, with rapid interconversion between these networks. Out of these conformations, a subset of specific arrangements of the atoms and hydrogen-bond interactions is required for the rate-determining reaction to proceed. Pronounced variations in water positions and hydrogen-bonded chains of water molecules, which are coupled to protein dynamics, have indeed been observed in classical molecular dynamics simulation of PSII in the time range of tens of nanoseconds[35]. This suggests a strong entropic contribution to the activation energy of any reaction that relies on a specific location of water molecules and hydrogen-bond interactions, as is the case for the $S_3' \rightarrow S_4$ transition shown in Fig. 3. The crystallographic analysis of semi-stable intermediates in the water oxidation cycle also supports reaction steps coupled to rearrangement of water molecules[25,36].

Our MEP calculations suggest protonation of the Asp61 carboxylate group in an electron transfer step that is coupled to a Grotthus-type movement of three protons, with the unconventional feature of a metal

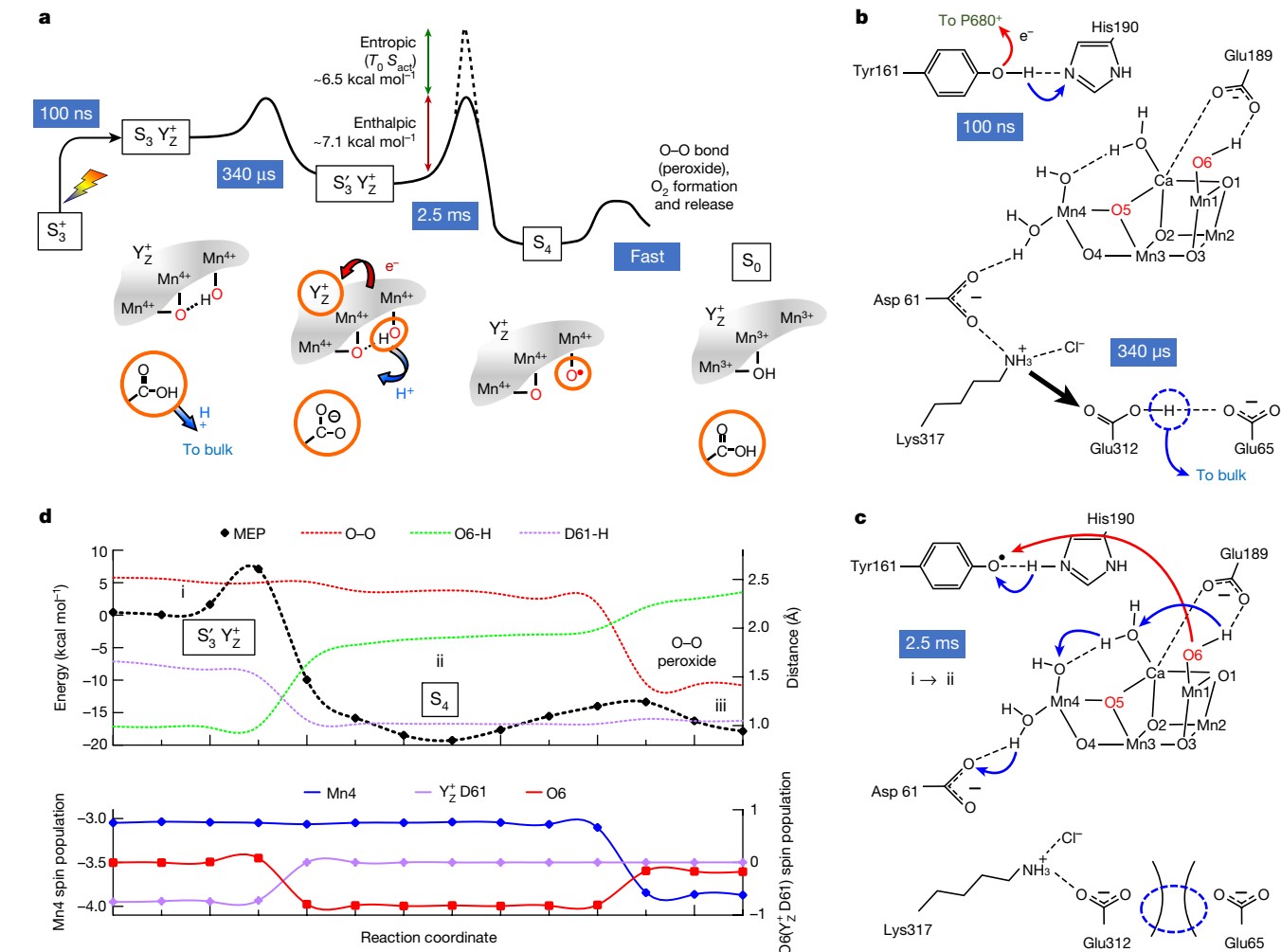

**Fig. 3 | Proton and electron transfer steps of the oxygen-evolution transition. a**, Schematic summary of experimental findings on reaction intermediates, time constants (reciprocal rate constants), enthalpic and entropic contribution of the activation energy (see Extended Data Fig. 5), and S-state assignment. The entropic contribution is the product of the entropy of activation ($S_{act}$) and the absolute temperature that corresponds to 20 °C ($T_0 = 293.15\,K$). Key features are highlighted with orange circles, the two oxygen atoms from 'substrate water' are shown in red, and charge-transfer events are indicated with arrows (red, electron transfer; blue, proton transfer). Out of the four Mn ions of the Mn$_4$Ca cluster, only the three Mn ions (Mn1, Mn3 and Mn4) that have accumulated oxidizing equivalents (holes) in preceding S-state transitions and are 'discharged' concomitantly with O$_2$ formation are

shown. The fourth hole transiently residing on the oxidized Tyr$_Z$ (denoted here as Y$_Z^+$) is filled by electron transfer from a substrate-water oxygen in the $S'_3 \rightarrow S_4$ transition. **b,c**, The rate-constant processes depicted in **a**, $S_3^+ \rightarrow S_3 \rightarrow S'_3$ (**b**) and $S'_3 \rightarrow S_4$ (**c**), are assigned to atomistic events, facilitated by computational results. The dotted blue circles highlight the creation of a proton vacancy induced by Tyr$_Z$ oxidation and activating Asp61 as a proton acceptor in the $S'_3 \rightarrow S_4$ transition via movement of the Lys317 sidechain. **d**, Energy values associated with MEP calculations and internuclear distances (top graph, dashed lines) as well as spin populations of the Mn ions, atoms and residues species (bottom graph, solid lines) that characterize the peroxide formation. See Extended Data Figs. 7 and 8, for structures complementing **d** and describing the complete oxygen-evolution transition.

ion (Mn4) acting as a relay. This single-electron–multi-proton step facilitates the critical step of Mn$^{IV}$-O$^\bullet$ formation (S$_4$) before O–O bond formation. The transient carboxylate deprotonation and reprotonation that we detected in the IR experiment might be assignable to the sidechain of Asp61. However, Asp61 is probably already deprotonated in the S$_3$ steady state of the Kok cycle[37–40] (Supplementary Information), possibly sharing a delocalized proton with the nearby Mn-bound water molecule[41]. In an alternative scenario, we identify a proton-carrying carboxylate 'dyad', Glu312 and Glu65, as the site of deprotonation within about 300 μs after the flash. Here we assume that Glu312 is initially protonated[42] (Supplementary Information) and transiently deprotonated during the oxygen-evolution transition. Other computational work suggests that Glu65 is initially protonated[43], but this difference is of minor importance. A shift of the nearby Lys317 sidechain could have a role in promoting Glu312–Glu65 deprotonation and increasing the proton affinity of Asp61, as shown in Extended Data Fig. 9

and Supplementary Information. Notably, Yano and coworkers[21] suggested that the Glu65–Glu312 pair serves as a proton-transport gate, on the basis of transient conformational changes during the S$_2 \rightarrow$ S$_3$ transition detected by serial crystallography. They did not investigate the oxygen-evolution transition itself. Furthermore, they could not demonstrate deprotonation of the carboxylate dyad. Nonetheless, assuming they are transferable to the S$_3 \rightarrow$ S$_4 \rightarrow$ S$_0$ transition, their finding of conformational changes of Glu65 provides support to our conclusion that Tyr$_Z$ oxidation induces residue rearrangements and Glu65–Glu312 deprotonation.

## Conclusions

Our study differs from previous studies in that it establishes a close connection between the sequence of events and reaction kinetics derived from experimental findings and those from computational analyses. We identify the rate-determining step in photosynthetic water

oxidation as the formation of a Mn(IV)-O$^{\bullet}$ radical, which corresponds to a one-electron oxidation of a substrate water molecule. The close coupling between electron transfer and Mn-OH deprotonation via Grotthus-type relocation of three protons facilitates the low enthalpic reaction barrier, with a computed barrier height that is as low as found experimentally. The concerted electron–proton reaction requires that the nuclei and hydrogen-bond pattern at the active site are perfectly arranged, which provides a plausible explanation for reaction slowdown by entropic factors. Computationally, we identify a low-energy path for the subsequent peroxide formation which is in line with a mechanistic idea pursued by Siegbahn and others[30–34,44], but we find that this is not the overall rate-determining step. However, we cannot completely exclude further low-energy routes towards O–O bond formation[40]—as they have also been discussed in relation to substrate–water exchange rates[45]—because an exhaustive search covering all conceivable reaction paths is unachievable. Previous investigations[30–34,40,44,46] did not consider the mechanistic features of a combined enthalpic–entropic barrier associated with the Mn-OH deprotonation and Mn(IV)-O$^{\bullet}$ radical formation, which we identify as the kinetically most demanding, overall bottleneck step in photosynthetic water oxidation.

The formation of Tyr$_Z^{ox}$, which provides the fourth electron hole at the active site, is insufficient to start the reactions leading to O$_2$ formation; the creation of a proton vacancy by the removal of a further proton from the active site is required[10]. Here we find that the proton is removed not from a substrate water molecule, but from a nearby base, which is identified as a carboxylate sidechain. This is the prerequisite for the subsequent concerted electron–proton transfer reaction that facilitates in the direct one-electron oxidation of a substrate water molecule, resulting in a Mn(IV)-O$^{\bullet}$ radical without formation of a Mn(V) intermediate. This last experimentally detectable state before the onset of water oxidation is thus formed in the presence of four accumulated electron holes (three Mn(III/IV) plus one Tyr$_Z^{ox}$) and an essential proton vacancy. Its characteristics might justify assignment to the S$_4$ state, as previously proposed[10]. However, this state is better classified as S$_3'$Tyr$_Z^{ox}$, to clarify the placement of the four electron holes and not to obscure the role of the tyrosine residue as an electron relay. Here we identify the S$_4$ state—whose identity had remained unknown since Bessel Kok developed the S-state cycle paradigm 50 years ago[1]—as an oxyl-radical state, with the four electron holes needed for O$_2$ formation available at the Mn$_4$Ca-oxo cluster itself (three Mn(III/IV) plus one O$^{\bullet}$). It is identified as a real reaction intermediate transiently formed before O–O bond formation—not a transition state, but experimentally undetectable for kinetic reasons.

An important development in photo- and electrocatalytic water oxidation for use in CO$_2$-neutral fuel production is the use of oxide materials based on earth-abundant metals (such as Mn, Fe, Co and Ni). These often share structural motifs with biological catalysts and frequently undergo metal-centred oxidation-state changes, which allows the accumulation of oxidizing equivalents before the onset of O$_2$ formation[47,48]—for example, by Mn(III/IV) oxidation[49]. Given our knowledge of photosynthetic water oxidation, we consider it plausible that in inorganic oxide materials, the formation of a reactive metal-O$^{\bullet}$ radical is the kinetically most demanding rate-determining step, which can proceed efficiently by coupled electron–proton transfer. Following the biological paragon, we expect that tuning the inorganic material for (1) extensive hole accumulation by metal oxidation at low overpotentials and (2) metal radical formation facilitated by proton-coupled electron transfer can lead to improved oxygen-evolution reaction catalyst materials based on earth-abundant resources.

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

# Methods

## Preparation of PSII particles

High-activity PSII membrane particles (>1,000 μmol $O_2$ per mg chlorophyll and hour) were prepared from spinach leaves as described[51]—with the Triton incubation time reduced[52] to 1 min—and stored at −80 °C in a betaine-rich cryo-buffer (1 M glycinebetaine, 25 mM 2-morpholin-4-yl ethanesulfonic acid (MES), 15 mM NaCl, 5 mM $MgCl_2$, 5 mM $CaCl_2$, pH 6.2 adjusted with NaOH) with a final chlorophyll concentration of about 2 mg ml$^{-1}$. Starting with 4 kg of spinach leaves, each PSII preparation yielded PSII membrane particles corresponding to about 200 mg chlorophyll. In total, about 40 kg of fresh spinach leaves were processed for this work, resulting in about PSII membrane particles corresponding to more than 1,500 mg of chlorophyll.

Before measurement, the PSII membrane particles were thawed, resuspended and washed twice by centrifugation at 50,000$g$ for 12 min in betaine-free variant of the cryo-buffer. The resulting pellet was mixed with an artificial electron acceptor (about 30 μl of a stock solution of 700 mM phenyl-p-benzoquinone (PPBQ) dissolved in dimethyl sulfoxide (DMSO)) yielding a final ratio of 7 μmol PPBQ per mg chlorophyll of the PSII membrane particles.

## Realization of 1,800 irradiation spots

The FTIR measurement cells consisted of two $CaF_2$ plates (diameter of 25 mm) sealed with vacuum grease. A small amount of PSII suspension with added electron acceptor sample was placed on one plate, with the second plate pressed firmly on top. The thickness of the sample was set using a self-made device using rotation and pneumatic pressure to press the plates together; simultaneous measurement of the absorbance at 1,650 cm$^{-1}$ allowed for the reasonably homogenous samples with an absorbance of 1 ± 0.2 absorbance units. Following thickness adjustment, two saturating laser flashes were applied (5-ns pulse width, 532 nm). These pre-flashes and the subsequent dark-relaxation period ensure a reproducible synchronization of the photosystems in the dark-stable $S_1$ state with the tyrosine residue denoted as $Tyr_D$ in its oxidized form[53].

Approximately 45 $CaF_2$ 'sandwiches', prepared as described above, were loaded into an automated sample change plate and mounted within a modified commercial FTIR spectrometer (Bruker Vertex 70, detector D317/BF) with an extended sample chamber and temperature control device, as shown in Extended Data Fig. 1 and detailed in the Supplementary Information. Concave focusing mirrors further focused the IR beam to a spot with a diameter of about 1.6 mm on the PSII sample. Each of the 45 $CaF_2$ sandwiches can have about 40 unique sample spots, resulting in about 1,800 individual spots per refill of the automated sample exchange system. Before measurement, the absorbance of each spot is sampled, and those not meeting the criteria of 1 ± 0.2 absorbance units are excluded from further measurement. This typically resulted in the exclusion of about 10% of the spots.

## Execution of the step-scan FTIR experiment

The temperature was regulated to 10 °C and a constant stream of dry air kept the sample chamber humidity below 2%. S-state cycling of the PSII membrane particles was achieved using saturating flashes from an excitation laser (5-ns flashes, 532 nm, Continuum Minilight II) coupled into the sample chamber through a transmission window. Spot size was adjusted using an iris to be larger than the diameter of the IR beam.

After loading the spectrometer and before start of the step-scan FTIR experiment, the samples were dark-adapted for at least 1 h. In the FTIR experiment, dark-adapted spots of the PSII sample were irradiated with 10 excitation laser flashes with a flash spacing of 700 ms. For each flash, the detector signal was recorded for 136 ms, with 24-bit analogue-to-digital conversion every 6 μs (sampling frequency of about 170 kHz), 6 ms before the flash and 130 ms after the flash, as illustrated by the timing diagram in Extended Data Fig. 1. Once data collection for

the 10th flash had been completed, the sample holder moved a 'fresh' sample spot into the IR beam path before initiation of the next flash sequence. At least 1 h of dark adaptation passed before the same sample spot was measured again. For the step-scan dataset, 334 interferogram mirror positions were repeatedly sampled, resulting in a spectral resolution of about 10 cm$^{-1}$. For signal averaging, about 230,000 measurement cycles where performed, meaning about 2,300,000 excitation flashes were applied and recorded within about 7 months. During this period, the experimental setup was repeatedly refilled with fresh PSII FTIR samples and data for total measurement time of about 60 days was recorded. After extensive assessment of the loss of functional stability over extended time periods, we decided to expose the FTIR PSII samples to laser flash sequences for typically 50–60 h (Supplementary Fig. 3). The primary data analysis as well as the correction for thermal artefacts are described in the Supplementary Information. Averaged time-resolved interferograms for all ten applied flashes and relevant Python analysis code has been made available[54,55].

For comparison, rapid-scan S-state difference spectra were recorded using the same type FTIR samples and automated sample exchange system as used for the step-scan data. The individual rapid-scan interferograms were collected within 37 ms, at a spectral resolution of 4 cm$^{-1}$. The steady-state spectrum was calculated by averaging spectra from 300 ms to 1 s after each laser flash.

## Decay-associated spectra

The time evolution of the obtained spectra was simulated with a standard exponential model:

$$F_v(t) = \sum_i A_{v,i} \cdot (1 - e^{-t/\tau_i}) + B_v, \qquad (1)$$

where $F_v(t)$ provides spectrum at a specific time, $t$, after the laser flash and $v$ indicates discrete wavenumbers spaced by 0.23 cm$^{-1}$. The time constants ($\tau_i$) and amplitude coefficients, $A_{v,i}$, were obtained by a simultaneous least-square fit of 2,582 transients corresponding to wavenumbers ranging from 1,800 cm$^{-1}$ to 1,200 cm$^{-1}$.

The amplitude coefficients, $A_{v,i}$, when plotted with respect to the wavenumber, represents the change in the recorded spectrum associated with time constant $\tau_i$, the DAS of the $\tau_i$ process. Note that according to equation (1), a positive value for $A_{v,i}$ represents a positive contribution the final steady-state spectrum. The offset $B_v$ represents the spectrum of all flash-induced changes that are faster than the time resolution of our experiment (sum of changes occurring within nanoseconds up to about 5 μs after the laser flash). Simulations begin 9 μs after the excitation flash. A Python toolbox written for this analysis has been made available[55].

## Acceptor-side correction

In the DAS of the $S_3 \rightarrow S_0$ transition, the amplitude of the millisecond phase is a combination (sum) of a minor contribution from the quinone reactions at the PSII acceptor side and major contribution of the oxygen-evolution step. For improved DAS of the oxygen-evolution step, we corrected for the quinone contribution based on the following rationale: the two-electron chemistry at the PSII acceptor side results in binary oscillation of the flash-number-dependent acceptor-side contribution, with one type of acceptor-side contributions dominating on odd flash numbers and a second type dominating on even flash numbers. This is supported by the clear similarity between the millisecond DAS of the $S_2 \rightarrow S_3$ and $S_0 \rightarrow S_1$ transitions (Supplementary Fig. 4). Consequently, we assume that the DAS of the millisecond phase in the $S_1 \rightarrow S_2$ transition also contributes to the millisecond DAS in the $S_3 \rightarrow S_0$ transition.

For the $S_1 \rightarrow S_2$ transition, DAS were obtained for 4 time constant components: 33 μs, 91 μs, 3.1 ms and 25 ms. The DAS of the 3.1 ms phase was subtracted from the corresponding DAS of the $S_3 \rightarrow S_0$ transition,

without further scaling. We note that the acceptor-side correction is uncritical to our conclusion, as demonstrated in Supplementary Fig. 5, but improves the precision of the DAS shown in Fig. 2.

## Activation energies from $O_2$ polarography

Thylakoid membranes from *T. elongatus* cells were prepared as described elsewhere[56]. After storage at −80 °C, the thylakoid suspension was thawed on ice for 60 min in complete darkness. After resuspension in an electrolyte buffer (150 mM NaCl, 25 mM MES, 1 M glycinebetaine, 5 mM $MgCl_2$, 5 mM $CaCl^2$, pH 6.2 adjusted with NaOH), an aliquot of 10 μl with a PSII concentration corresponding to 10 μg of chlorophyll was placed into the cavity of a centrifugable two-electrode assembly followed by centrifugation in a swingout rotor at 10,000*g*. We used a custom-made centrifugable static ring-disk electrode assembly of a bare platinum and silver-ring electrodes to perform time-resolved oxygen polarography measurements[56,57]. A custom-made potentiostat provided the polarization voltage (−0.95 V at the Pt electrode versus the Ag ring electrode) for $O_2$-reduction at the bare Pt electrode, which was switched-on 15 s before the first excitation flash. The current signal was recorded for 500 ms (20 ms before and 480 ms after each flash, for 80 flashes with 900 ms spacing) using a first-order high-pass filter (time constant of 100 ms) for suppression of slow drift contributions in the current signal. The S-state transitions were induced by saturating flashes of red light (613 nm, 40 μs of flash duration). The light source was a high-power light-emitting diode (LED) operated at a maximum current density of around 150 A, as facilitated by capacitor discharge. The temperature during data acquisition was set using Peltier elements and monitored by a miniature temperature sensor immersed in the sample buffer.

For accurate (correct) and precise determination of the $O_2$ formation rate constant, the recorded current transients were simulated by numerical integration of the one-dimensional $O_2$ diffusion equation, involving light-induced $O_2$ production within the PSII layer covering the electrode (source terms), $O_2$-consumption at the electrode (sink terms), and accounting for the high-pass filter characteristics (see refs. 56,57, software developed by I. Zaharieva). By variation of model parameters, the recorded $O_2$ transients were simulated until optimal agreement between experimental and simulated transients was reached (least-squares curve fitting).

After determination of the activation energy ($E_{act}$) and pre-exponential factor ($A$) following the classical approach of Arrhenius, the enthalpy of activation ($H_{act}$) and entropy of activation ($S_{act}$) were determined using the Eyring equation of transition-state theory[58] (also called Eyring–Polanyi equation) with a transmission coefficient of unity, analogous to the treatment in refs. 59,60.

In Extended Data Fig. 5e, at each temperature the data points (filled circles) indicate the average of three values of the time constant of oxygen evolution ($\tau_{ox}$) obtained by simulation of the $O_2$ transients of three independent experiments with averaging of 70–80 flash-induced $O_2$ transients per experiment; the error bars indicate the corresponding standard deviation ($n = 3$). The value of $E_{act}$ and its uncertainty range result from determination of the slope of the regression line for ln ($\tau_{ox}$) versus ($k_B T)^{-1}$ (shown as a dotted line; $k_B$, Boltzmann constant; $T$, temperature in Kelvin) without weighting by standard deviations; the provided uncertainty range of $E_{act}$ corresponds to the 1σ confidence interval of the slope value. The uncertainty ranges for $H_{act}$ and $S_{act}$ were estimated such that their relative uncertainty corresponds to the relative uncertainty in $E_{act}$ resulting in ranges of about ±9 meV for $H_{act}$ and $T_0 S_{act}$. Alternatively, the uncertainty ranges could be estimated by assuming the same absolute uncertainty in $E_{act}$, $H_{act}$ and $T_0 S_{act}$, which would result in uncertainty ranges of about ±10 meV for both $H_{act}$ and $T_0 S_{act}$.

## QM/MM molecular dynamics simulations

The model used for quantum mechanics/molecular mechanics (QM/MM) calculations is the same adopted in previous studies[26,28]. The model consists of the D1, D2 and CP43 protein domains, the respective cofactors, the $Mn_4Ca$ cluster, and water molecules present in this region. (D1, D2 and CP43 indicate the protein subunits of PSII, which correspond to the *PsbA*, *PsbB* and *PsbC* gene products.) The $Mn_4Ca$ cluster, with its ligands present in the first shell (D1-Asp170, D1-Glu189, D1-His332, D1-Glu333, D1-Asp342, D1-Ala344 and CP43-Glu354), plus additional residues in the second shell (D1-Asp61, D1-Tyr161, D1-His190, D1-His337, D1-Asn181, D1-Ile60, D1-Ser169 and CP43-Arg357) were treated at the DFT level. Additionally, the first 14 water molecules closest or directly coordinated to the $Mn_4Ca$ cluster, and the chloride anion close to Glu333 were also treated at DFT level. See Supplementary Fig. 18 for graphical representation of the DFT region. The rest of the system was treated at classical level using AMBER99SB force field[61] to describe the protein residues and the general AMBER force field (GAFF)[62] for the description of the other cofactors present in the investigated region of PSII.

QM/MM calculations reported in this study have been carried out using the CP2K package[63]. QM/MM molecular dynamics simulations were performed in the NVT ensemble using a Nose–Hoover thermostat[64,65] (time constant $\tau = 0.1$ ps) to couple the system with a thermal bath at $T = 298.15$ K. A cutoff for the plane-wave expansion of 320 Rydberg was used to treat the quantum region with a cubic cell 28.0 × 28.0 × 28.0 A. The PBE+U scheme[66] was employed using the DZVPMOLOPT-SR-GTH Gaussian basis set optimized for molecular systems[67]. Electrostatic coupling between the classical and quantum regions of the system was treated by means of fast Gaussian expansion of the electrostatic potential[63]. A time step of 0.5 fs was used.

To study the effect of the protonation of Asp61 by W1 water molecule, we applied a position restraint on the proton between Asp61 and W1 to force the protonation of Asp61. The O-H equilibrium distance for the restraint was set to 1 A with a force constant of 0.01 internal units. The restraint has been applied for a limited amount of time and the results are described in the Supplementary Information.

## MEP calculations

MEP calculations have been performed on a PSII gas-phase model directly extracted from the QM-treated region of the QM/MM simulations described in ref. 26. The B3LYP[68,69] functional with TZVP-MOLOPT-SR-GTH Gaussian basis set has been employed for all the gas-phase calculations. A 28.0 Å side cubic cell and a cutoff for the plane-wave expansion of 320 Rydberg was used to treat the quantum region. The systems have been simulated following the 'high oxidation-state paradigm' of the $S_3$ state, consistent with previous computational work[26,30,33,40,70,71] and in line with most experimental analyses, as reviewed in ref. 72. The 'high oxidation-state paradigm' in the $S_3$ state corresponds to an oxidation pattern of the four Mn ions equal to IV, IV, IV and IV. The spin multiplicity, $M$, equals 6 with a spin moment of 5/2 as determined experimentally[73].

The rationale of the MEP calculations and the specific nudged elastic band (NEB) approach is illustrated in Supplementary Fig. 19. To calculate the MEP, it is necessary to provide the algorithm with an initial and a final structure of the reaction, and with a preliminary set of intermediate structures, representing the reaction coordinate, denoted herein as 'replicas'. The starting and final structures have been extrapolated by QM/MM molecular dynamics simulations reported in previous work[26,28] involving different redox states of the Mn cluster and Tyr$_Z$. The dynamic properties of the same QM region (220 atoms; Supplementary Fig. 18), both in the $S_3$ state and after Tyr$_Z$ oxidation were sampled for at least 10 ps. The MM system included 37,000 atoms for all the simulated systems. Extracting the initial and final structures of the MEP calculation from QM/MM molecular dynamics simulations at finite temperature supports finding the optimal hydrogen-bond network for the water and the residues included in the simulated system. Still, we cannot exclude that an even more favourable hydrogen-bond network could be reached, for example increasing the simulation time of the QM/MM molecular dynamics trajectory.

To properly represent all the intermediates and transition states characterizing the reaction coordinate we used 14 different replicas. Three of the 14 structures, the intermediates i, ii, and iii in Extended Data Fig. 7, were extracted from previous QM/MM molecular dynamics simulations. The remaining initial structures were obtained by linear interpolation of the cartesian coordinates between i and ii geometries and between ii and iii geometries. The MEP algorithm is meant to optimize simultaneously all these replicas such that the corresponding path satisfy the requirement of an MEP. For the first half of the MEP, from intermediate i to ii (proton relocation and O6 oxidation), the first and the last structures of the initial-guess structure were extracted from the QM/MM molecular dynamics simulation reported in ref. 28. For the second portion of the reaction pathway (peroxide formation), the structure of the final point (iii) has been extracted from the QM/MM molecular dynamics simulation reported in ref. 26. The MEP has been calculated using the NEB method as implemented in CP2K package[63,67,74]. The algorithm form of NEB calculation applied to our model is the improved tangent version (IT-NEB)[75]. This setup has been previously applied successfully in the study of water delivery and oxygen reorganization in the $S_2$ to $S_3$ transition of Kok cycle[70,71].

After convergence, the calculated energies of the 14 geometries correspond to the internal energy variation of the system along the reaction pathway (all the geometries are reported in Supplementary Fig. 20 and provided as Supplementary Data). In the MEP calculations the calculated energy is the variation of internal energy. We note that for the proton shuffling and peroxide formation in protein the variation of volume and pressure is expected to be negligible (PV component), therefore, the internal energy variation ($\Delta U$) is a good approximation of the enthalpy variation ($\Delta H$).

The internal energy difference between a specific intermediate state (i or ii) and the related transition state is therefore a good estimate of the respective activation enthalpy ($\Delta H^*1$; $\Delta H^*2$). The difference of internal energy between two intermediates (between two of the three states i, ii, and iii) corresponds to the respective enthalpy variation ($\Delta H1$; $\Delta H2$). All the values are reported in Extended Data Fig. 7 and its caption.

### Classical molecular dynamics simulations

Two classical molecular dynamics simulations have been performed to investigate the possible conformations adopted by D2-Lys317. The first simulation (simulation 1) was carried out considering both D1-Asp61 and D2-Glu312 as deprotonated. In the second simulation (simulation 2) Glu312 of the D2 subunit was set as protonated, while D1-Asp61 was still deprotonated. Apart from this difference, the two simulations are based on the same setup described in ref. 76. Protein residues were described using the AMBER99SB force field[62] while other molecules present in the crystal structure (that is, β-carotene, chlorophyll $a$, pheophytin $a$, plastoquinone 9, heptyl 1-thiohexopyranoside, dodecyl-β-D-maltoside and the four lipids digalactosyl diacylglycerol (DGDG), monogalactosyl diacylglycerol (MGDG), phosphatidylglycerol (PG) and sulfoquinovosyl diacylglycerol (SQDG) have been described by the generalized Amber force field (GAFF)[62]. Optimization and electrostatic potential analysis were performed by using Gaussian 03 at the Hartree–Fock level with the 6–31G* basis set. The protonation state of the histidines bound to haem molecules, or directly interacting with iron atoms or the magnesium atom in chlorophylls was chosen accordingly with their relative orientation in the X-ray structure. The other titratable residues were considered in their standard protonation state with the only exception of Glu312 of the D2 subunit in simulation 2 considered protonated. The PSII structure was embedded into a membrane bilayer composed by dioleoylphosphatidylcholine (DOPC) lipids described by the GAFF-based force field developed by Siu et al.[77]. The system was then solvated in a box with dimensions 27.0 × 16.3 × 14.2 nm using TIP3p water model[78]. Molecular dynamics simulations were performed using the GROMACS software package[79]. Long-range electrostatic interactions were calculated using particle mesh Ewald method[80],

with a grid spacing of 0.12 nm and a short-range cutoff of 1.0 nm. The LINCS algorithm[81] was applied to constrain the bond lengths of the hydrogen atoms to a constant value. A time step of 2 fs was used for numerical integration of the equations of motion. The temperature was kept constant by coupling the system to a Nose–Hoover thermostat (298 K) with a coupling time constant[64,65] $\tau_T = 0.1$ ps. The system was also weakly coupled to a pressure bath (1 bar) with a coupling time constant $\tau_P = 1.0$ ps, using Parrinello–Rahman barostat[82,83]. The final system was composed by almost 650,000 atoms. The two molecular dynamics simulations were carried out for 50 ns in NPT ensemble.

### Reporting summary

Further information on research design is available in the Nature Portfolio Reporting Summary linked to this article.

### Data availability

The complete step-scan data obtained for 230.000 sequences of 10 laser flashes applied to dark-adapted PSII is available in form of averaged time courses of the detector signal for all 334 mirror positions and 10 exciting laser flashes per mirror position (at Zenodo: https://doi.org/10.5281/zenodo.7681840). We furthermore provide the spectrum used for correction of the heat artefact at the same location. The coordinates of the 14 MEP structures are provided as a Supplementary Data file in standard PDB format. Source data are provided with this paper.

### Code availability

Two short Python toolboxes used to correct time-resolved Fourier transform data of high-activity PSII membrane particles from spinach for starting population and cycle inefficiency (miss factor) and to perform a global fit of a dataset to generate DAS are available at Zenodo (https://doi.org/10.5281/zenodo.7682034).

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

**Acknowledgements** The FTIR step-scan experiment on PSII was developed by B. Süss, M. Görlin and P. Chernev; we thank them for their basic and indispensable contributions to development of the experiment. The authors thank Y. Zilliges, Z. R. Pérez and J. Hantke for photobioreactor operation (for growth of *T. elongatus*) and preparation of the cyanobacterial thylakoid membranes; and R. Burnap and M. Haumann for stimulating discussions and critical reading of early manuscript versions. P.G., M.S., R.A. and H.D. gratefully acknowledge financial support by the Deutsche Forschungsgemeinschaft (DFG, German Research Foundation) provided to the collaborative research centre on Protonation Dynamics in Protein Function (SFB 1078, project A4/Dau) and under Germany's Excellence Strategy–EXC 2008/1–390540038–UniSysCat. M.C., D.N. and L.G. are grateful for the computational resources provided by the Partnership for the Advanced Computing in Europe (PRACE, project Pra14_3574) and CINECA computer centre (project IsC37_QMMMS3S4).

**Author contributions** H.D. and L.G. designed the experimental and computational research, respectively. M.S. optimized the FTIR step-scan experiments and collected the data. P.G. analysed the FTIR data. R.A. collected and analysed the $O_2$ polarography data. M.C. and D.N. performed the computational research. H.D., L.G. and P.G. wrote the paper.

**Funding** Open access funding provided by Freie Universität Berlin.

**Competing interests** The authors declare no competing interests.

**Additional information**
**Correspondence and requests for materials** should be addressed to Leonardo Guidoni or Holger Dau.

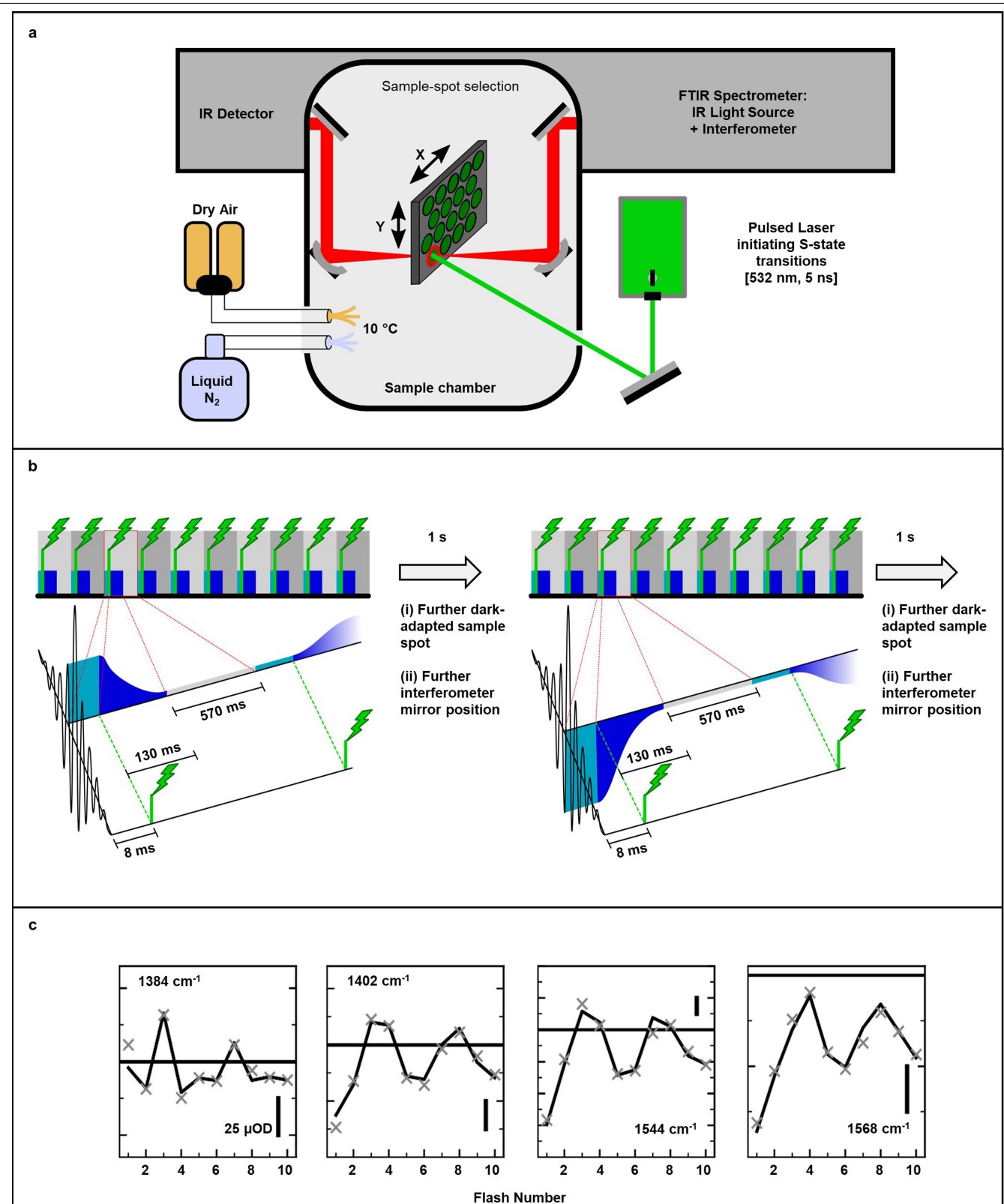

**Extended Data Fig. 1** | See next page for caption.

**Extended Data Fig. 1 | Experimental setup, timing scheme, and infrared data confirming synchronized reaction-cycle advancement. a**. Schematic representation of the experimental setup. A standard FTIR instrument was modified with an extended, air-tight sample chamber to harbor the automated sample exchange system and combined with a pulsed nanosecond laser. After removal of water vapor from ambient air (dry air generator), the dry gas stream flooded the sample changer. Within the sample changer the temperature was kept a constant 10 °C by a flow of cold nitrogen gas. Stepping motors moving the sample state in x-y direction facilitated laser-flash excitation (and data collection) at 1800 spots of dark-adapted PSII samples. **b**. Timing scheme of the experiment. At each sample spot, 10 sequential flashes were applied while the interferometer had been set to a specific mirror position. The IR detector recorded 8 ms before each flash, 130 ms after each flash and waited 570 ms between the flashes without recording data. Once the detector had finished recording after the tenth flash, the sample changer moved to a 'fresh' dark-adapted sample spot and the interferometer mirror moved to a new position within about 1 s. The 10-flash sequence was applied again at the new positions and the whole timing sequence was repeated numerous times. **c**. By application of 10 sequential laser flashes, the PSII can cycle up to 2.5 times through its S-state cycle. The synchronized advancement in the S-state cycle is verified by a period-of-four pattern in the infrared absorption changes (here averaged from 50–130 ms after the laser flash for 4 selected wavenumbers). Solid lines show the data points and the grey crosses represent a simulation using the deconvolved S-States with the derived miss factor and starting populations. Scale bars correspond to 25 μOD.

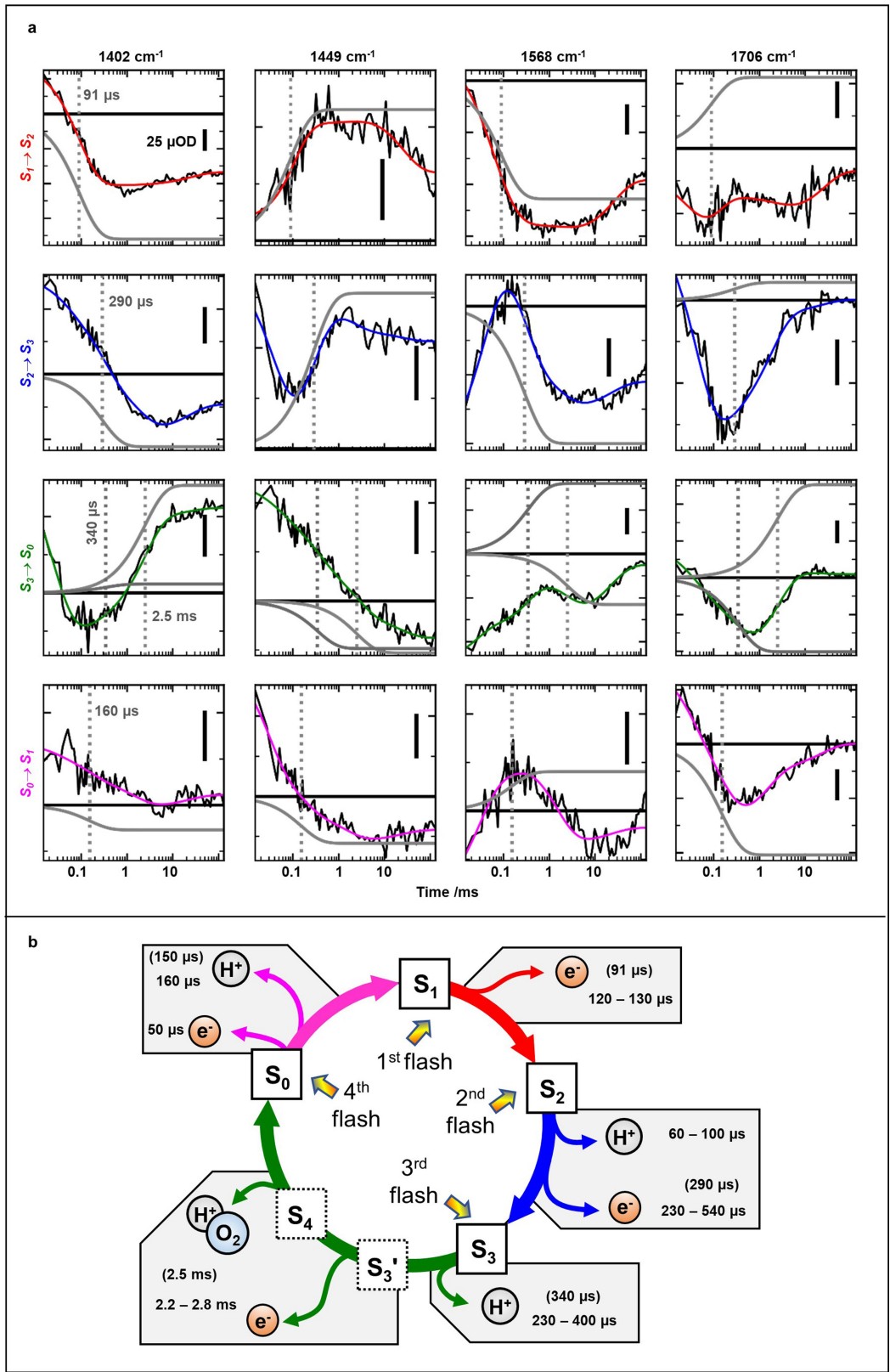

**Extended Data Fig. 2 | Select infrared transients demonstrating that previously identified rate constants are resolved also in the step-scan FTIR experiment. a.** Transients at select wavenumbers for the four deconvolved S-state transitions. These transients are shown to demonstrate that previously identified reaction kinetics are recoverable in this work. Smooth colored lines represent multi-exponential simulation (least-square fit) of the IR transients. Grey dashed lines mark the identified time constants while grey transients show the respective simulated contributions to the total transient. In each panel, the respective pre-flash level is indicated by a black line. The complete set of simulation parameters can be found in Supplementary Table 2. **b.** S-state cycle with previously determined time constants values (see Supplementary Table 3) and those found in this work (in parenthesis).

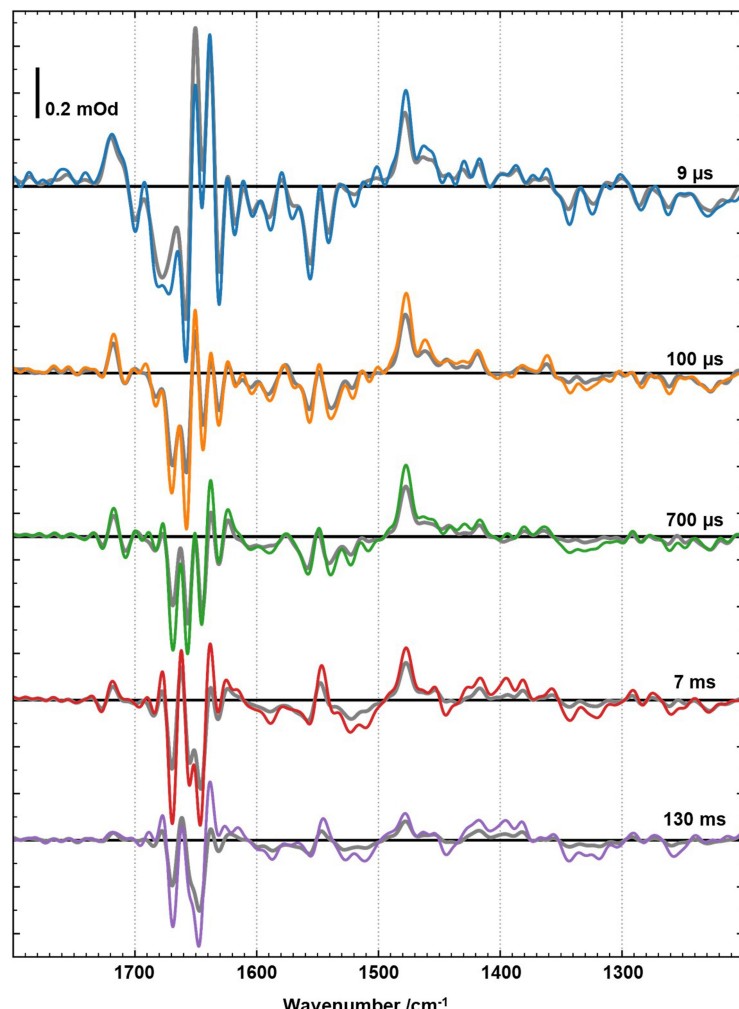

**Extended Data Fig. 3 | Infrared spectra at various times after application of the laser that initiates the oxygen-evolution transition.** The grey line shows the spectral changes induced by the 3rd laser flash applied to dark-adapted PSII. The colored lines are corrected (deconvolved) for imperfect advancement in the S-state cycle as detailed in the Supplementary Material.

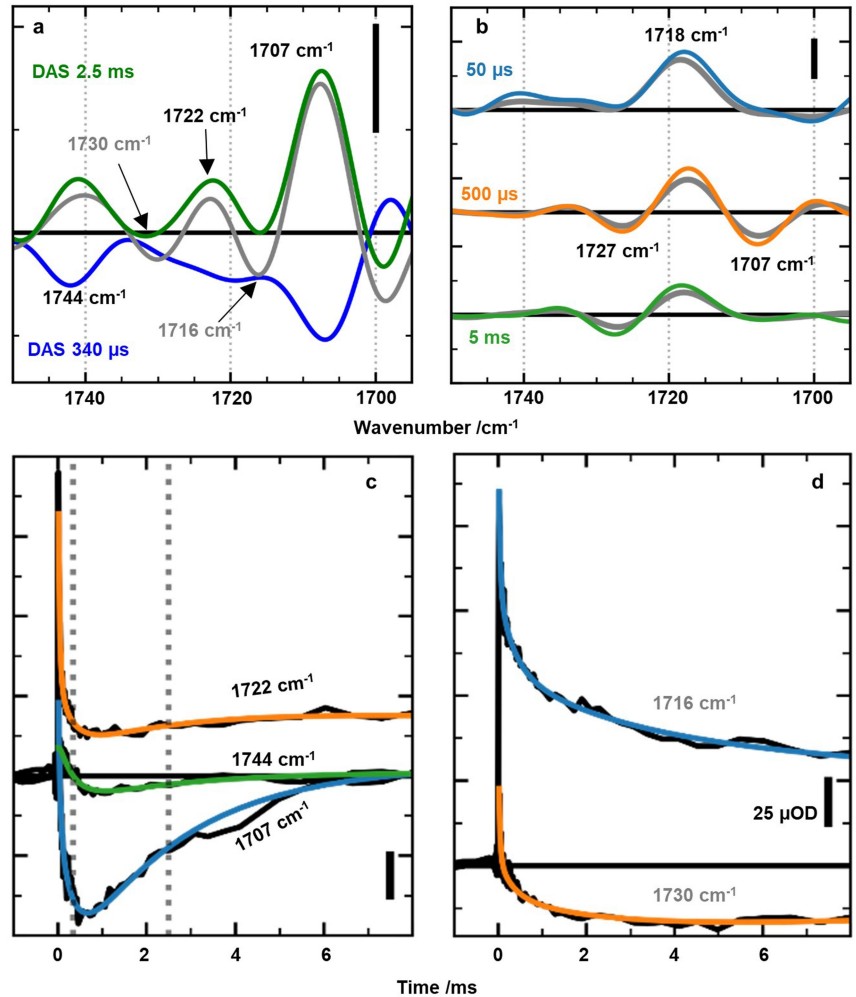

**Extended Data Fig. 4 | Carbonyl band spectra and IR transients between 1695 and 1750 cm⁻¹ for the oxygen-evolution transition (S₃->S₄->S₀). a.** Decay-associated spectra of the 340 µs component and of the 2.5 ms component, the latter before (grey line) and after (green line) correction for acceptor side contributions. The acceptor side correction results in three positive peaks for the 2.5 ms component, indicating that band shift cannot explain these three peaks at 1730 cm⁻¹, 1722 cm⁻¹ and 1707 cm⁻¹. **b.** Spectra at selected times for the oxygen-evolution transition induced by the 3rd laser flash before (grey lines) and after (colored lines) deconvolution. The high level of similarity indicates that in this spectral region the deconvolution correction is uncritical as it hardly modifies the 3rd-flash data. **c.** Transients displaying reversible behaviour of the 340 µs and 2.5 ms phases reflecting carboxylate deprotonation and reprotonation. **d.** Transient changes assignable to acceptor side contributions. All scale bars correspond to 25 µOD. For details on correction for acceptor side contributions and S-state deconvolution, see Supplementary Information.

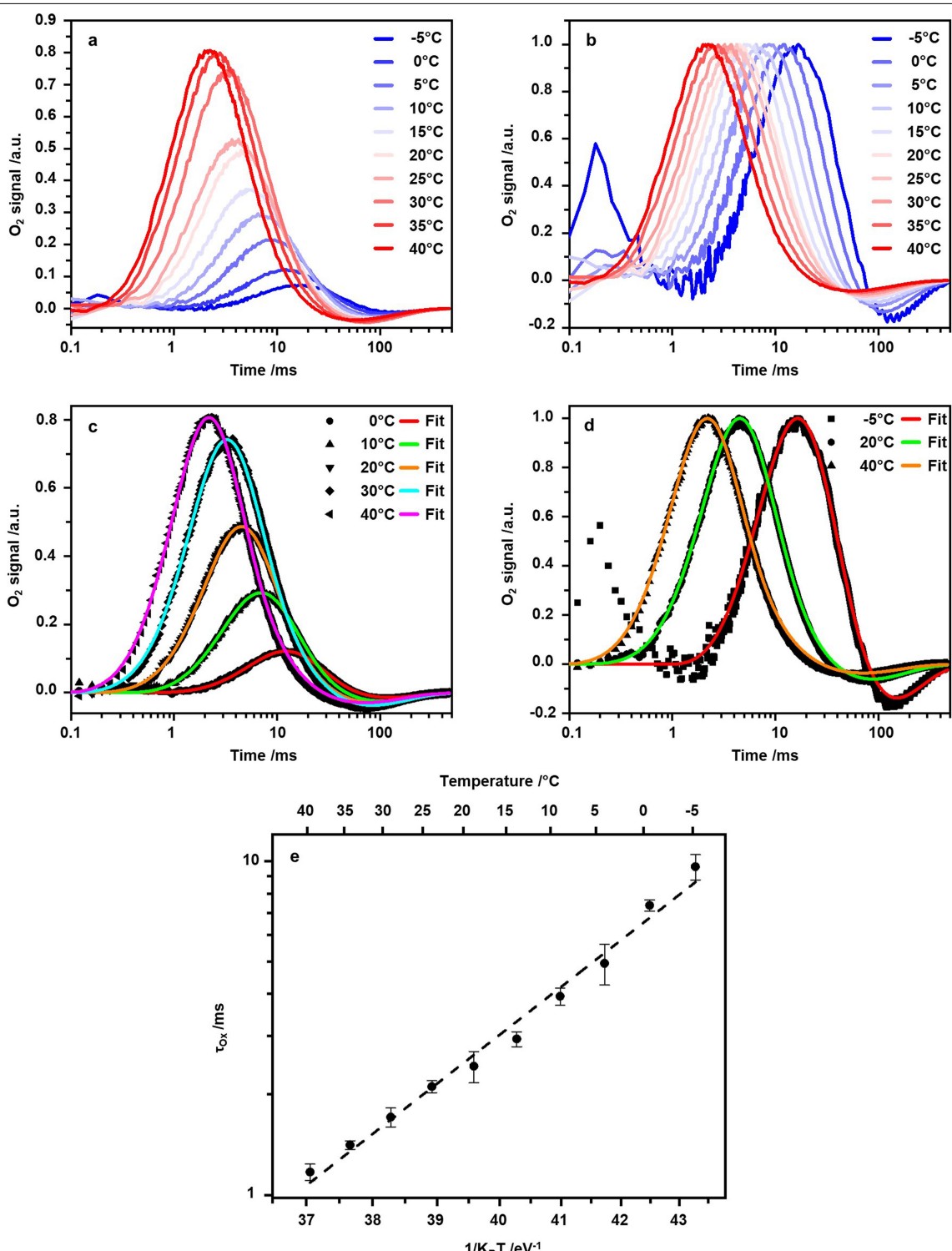

**Extended Data Fig. 5** | See next page for caption.

**Extended Data Fig. 5 | Time-resolved $O_2$-polarography and determination of activation energy of the $O_2$-formation step for cyanobacterial PSII from *T. elongatus*.** The time courses (transients) of $O_2$-evolution were (i) measured by time-resolved $O_2$-polarography for a PSII layer deposited by centrifugation on a bare platinum electrode and (ii) simulated on grounds of a physical diffusion model, including a least-square fit of the simulation parameters (see Methods section). The complete set of simulation parameters is provided in Supplementary Fig. 6. **a**. $O_2$-evolution transients at temperatures ranging from −5 °C to +40 °C. Each transient was obtained by averaging all the $O_2$-transients induced by 230 flashes of visible light. **b**. The same transients as shown before but normalized at the respective peak value. In **c** and **d**, selected transients are shown (black symbols) with their respective fit results (colored lines), either with original amplitudes (in c) or normalized to unity (in d). **e**. Arrhenius plot of $\tau_{ox}$, the time constant (reciprocal rate constant) of the oxygen evolution reaction. This plot delivers an Arrhenius activation energy, $E_{act}$, of 335+/−10 meV (7.73+/−0.23 kcal/mol) with a pre-exponential frequency factor, $A$, of $2.2 \cdot 10^8 \, s^{-1}$. After determination of activation energy and pre-exponential factor following the classical approach of Arrhenius, the enthalpy of activation ($H_{act}$), entropy of activation ($S_{act}$) were determined using the Eyring equation of transition-state theory (also called Eyring-Polanyi equation) with a transmission coefficient of unity, resulting in values of 310+/−9 meV (7.15+/−0.21 kcal/mol) for $H_{act}$ and of 284+/−9 meV (6.55+/−0.21 kcal/mol) for $T_0 S_{act}$ (with $T_0 = 20$ °C). The applicability of transitions state theory is discussed in Supplementary Information SII.8. The error bars provide the standard deviation of the mean value calculated for three independent experiments at each temperature; the confidence intervals of the energy values are derived from the probable error in the slope of the regression line (broken line).

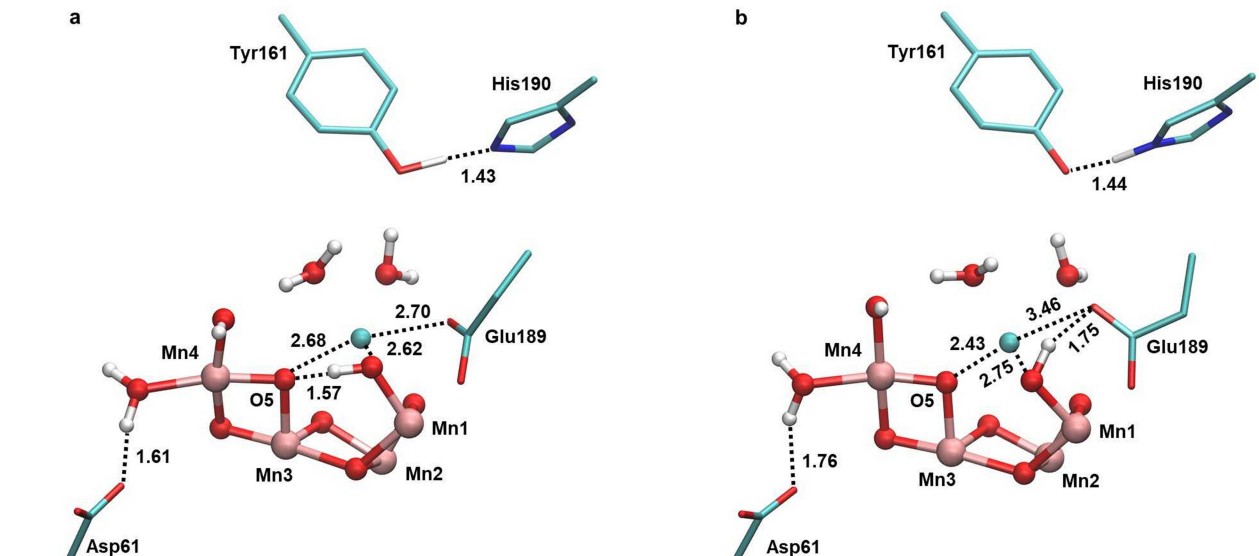

**Extended Data Fig. 6 | The two most stable hydrogen-bonds of protonated O6 (in both S₃ and S₃' state).** Upon oxidation of Tyr$_Z$ (Tyr161), the relative stability of the two reported conformers is reversed, conformer-**a** becoming more stable than conformer-**b**, thus favouring the oxyl radical formation as described elsewhere[26]. In both cases, the deprotonated Asp61 is stably interacting with the W1 water molecule. Relevant distances are highlighted by thin lines and reported in both panels for comparison. Manganese atoms are shown in pink, calcium as cyan sphere, oxygen in red, carbon in cyan, and hydrogen in white.

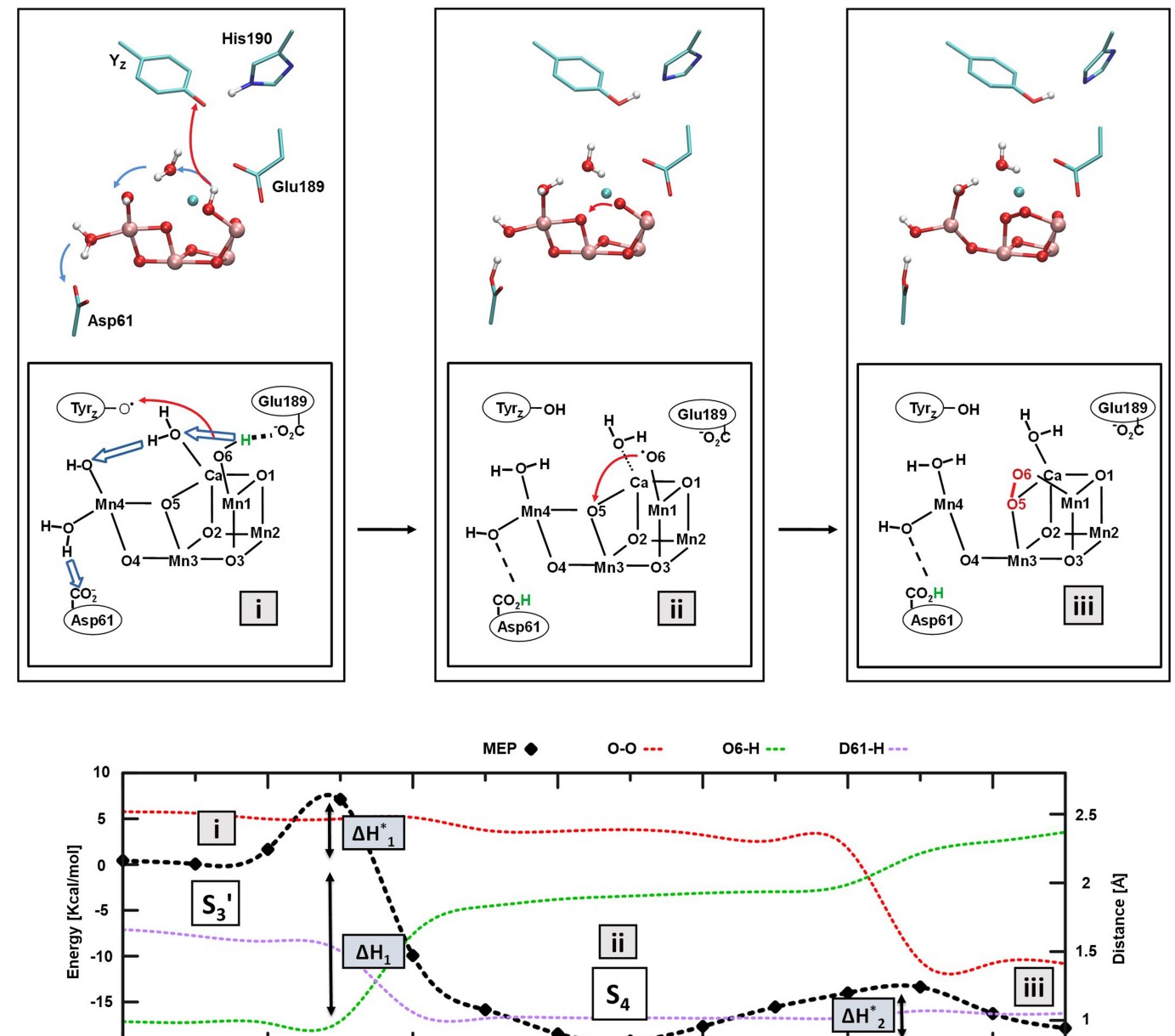

**Extended Data Fig. 7 | Atomic and electronic rearrangements leading to peroxide formation.** The top panels show molecular sketches of the atomic and electronic motions associated with the energy barriers to overcome along the peroxide formation reaction shown in the bottom panel. The values for the indicated enthalpies of activation and stabilization are: $\Delta H^*_1$, +7.0 kcal/mol; $\Delta H^*_2$, +5.8 kcal/mol; $\Delta H_1$, −19 kcal/mol; $\Delta H_2$, +1 kcal/mol.

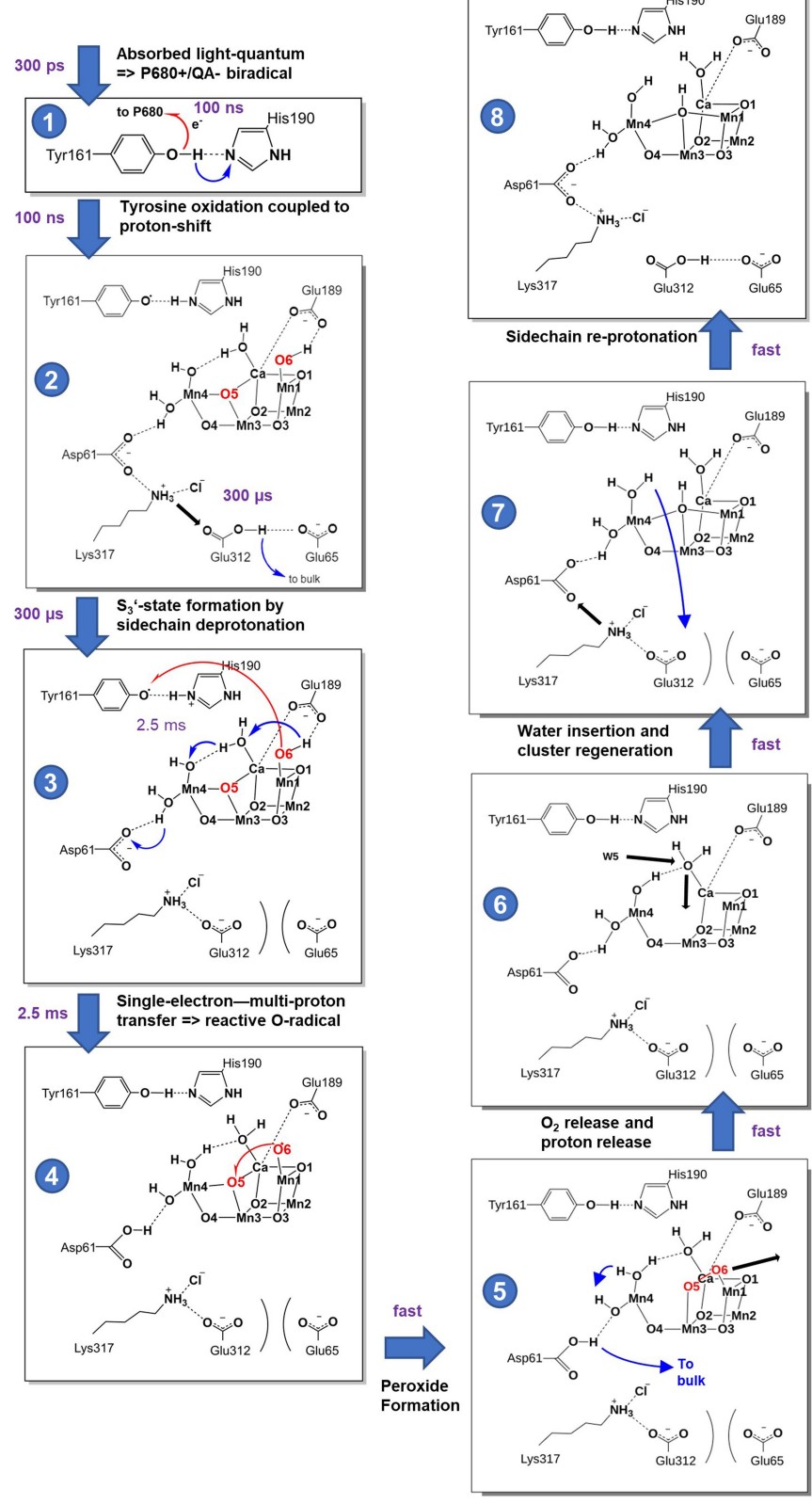

**Extended Data Fig. 8** | See next page for caption.

**Extended Data Fig. 8 | Complete picture of events in the oxygen-evolution transition.** Proton and electron transfers are highlighted by blue and red arrows, whereas relocations of heavy atoms are indicated by black arrows. Substrate oxygen atoms (O5 and O6) are depicted in red. The mechanism of the steps 1–5 was investigated in the present study, while the events occurring between the steps 5 and 7 (i.e. the release of molecular oxygen after the peroxide bond formation and the $Mn_4CaO_5$ cluster restoration) were described in ref. 84. The events in the transition from **7** to **8** are plausible, but currently not backed up by calculations. The eight panels illustrate the following sequence of events: Oxidation of Tyr161 in the $S_3$ state, which is coupled to the proton transfer from Tyr161 to His190 (**1**), induces a conformational change involving the side chain of Lys317, resulting in the approach of Lys317 to Glu312 and deprotonation of Glus312 (**2**) within about 340 µs after the laser flash. Within the next 2.5 ms, the transfer of one electron from the O6 atom to Tyr161 coupled to a concerted Grotthus-type relocation of three protons (**3**), resulting in $S_4$ formation by radicalization of O6 coupled to protonation of Asp61 (**4**). Thereafter, the O6 radical forms a peroxide bond with the oxygen atom O5 (**5**) and the subsequent deprotonation of Asp61 and release of molecular oxygen. The vacancy site formed by the oxygen evolution step is rapidly refilled with a water molecule coordinated to the $Ca^{2+}$ ion, with simultaneous proton transfer to the hydroxide ion bound to Mn4, and insertion of W5 into the coordination sphere of $Ca^{2+}$ (**6**). The restoration of the $Mn_4CaO_5$ in the $S_0$ state is completed by the deprotonation of a water molecule coordinated to Mn4, relocation of Lys317 close to Asp61, and protonation of Glu312 (**7** and **8**).

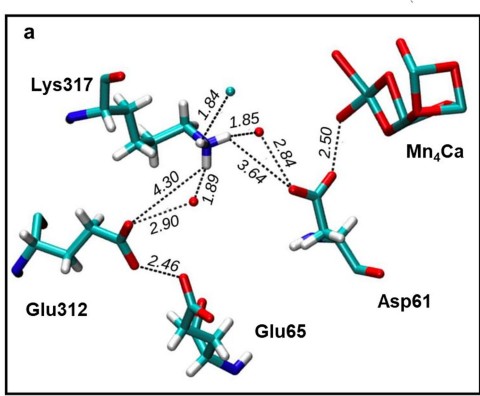

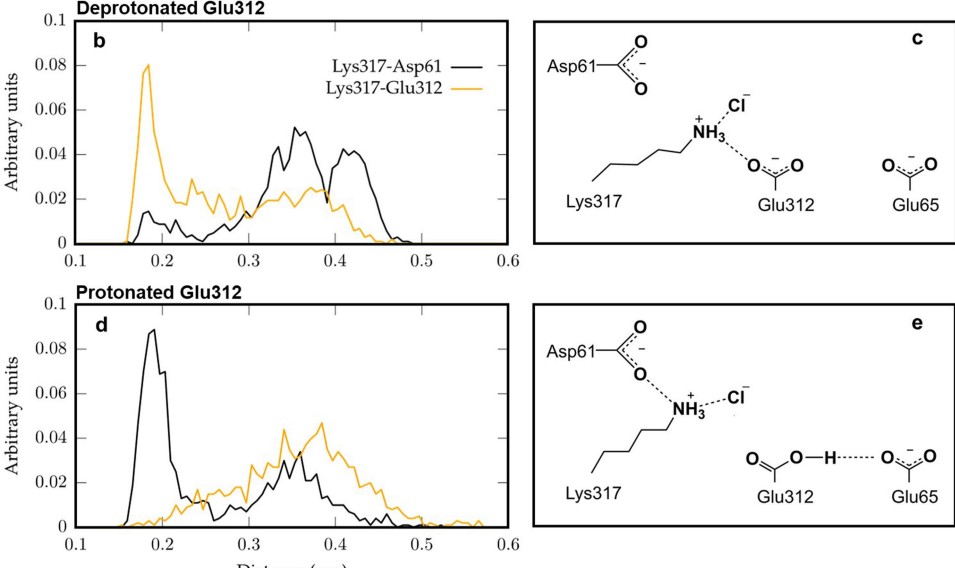

**Extended Data Fig. 9 | Conformations sampled by D1-Asp61, D2-Glu312, and D2-Lys317. a**. Sticks representation of the $Mn_4Ca$ cluster and neighboring residues (Asp61, Glu65, Glu312 and Lys317) for the two-flash state ($S_3$ state) of Photosystem II from the crystallographic model of Kern et al. (PDB ID: 6DHO). A $Cl^-$ ion as well as two water oxygens are also shown as balls. **b**. Distributions of distances between Lys317 and the two residues Asp61 and Glu312 sampled along 50 ns of classical MD simulation with both Asp61 and Glu312 deprotonated (simulation for $S_1$ geometry). **c**. Scheme of a representative configuration of

Asp61, Glu65, Glu312 and Lys317 sampled in the $S_1$ simulation. In this configuration Lys317 is in close contact with deprotonated Glu312. **d**. Distributions of distances between Lys317 and the two residues Asp61 and Glu312 sampled along 50 ns of classical MD simulation with deprotonated Asp61 and protonated Glu312 (simulation for $S_2$). **e**. Scheme of a representative configuration of Asp61, Glu65, Glu312 and Lys317 sampled in the $S_2$ simulation. Lys317 strongly interacts with Asp61, while protonated Glu312 interacts with Glu65.

# Reporting Summary

## Statistics

For all statistical analyses, confirm that the following items are present in the figure legend, table legend, main text, or Methods section.

| n/a | Confirmed | |
|---|---|---|
| ☐ | ☒ | The exact sample size (*n*) for each experimental group/condition, given as a discrete number and unit of measurement |
| ☒ | ☐ | A statement on whether measurements were taken from distinct samples or whether the same sample was measured repeatedly |
| ☒ | ☐ | The statistical test(s) used AND whether they are one- or two-sided <br> *Only common tests should be described solely by name; describe more complex techniques in the Methods section.* |
| ☒ | ☐ | A description of all covariates tested |
| ☒ | ☐ | A description of any assumptions or corrections, such as tests of normality and adjustment for multiple comparisons |
| ☐ | ☒ | A full description of the statistical parameters including central tendency (e.g. means) or other basic estimates (e.g. regression coefficient) AND variation (e.g. standard deviation) or associated estimates of uncertainty (e.g. confidence intervals) |
| ☒ | ☐ | For null hypothesis testing, the test statistic (e.g. *F*, *t*, *r*) with confidence intervals, effect sizes, degrees of freedom and *P* value noted <br> *Give P values as exact values whenever suitable.* |
| ☒ | ☐ | For Bayesian analysis, information on the choice of priors and Markov chain Monte Carlo settings |
| ☒ | ☐ | For hierarchical and complex designs, identification of the appropriate level for tests and full reporting of outcomes |
| ☒ | ☐ | Estimates of effect sizes (e.g. Cohen's *d*, Pearson's *r*), indicating how they were calculated |

*Our web collection on statistics for biologists contains articles on many of the points above.*

## Software and code

Policy information about availability of computer code

| Data collection | Only instrument-specific code was used that is irrelevant for our results and conclusions. |
|---|---|
| Data analysis | Two short Python toolboxes were used to correct time-resolved Fourier transform data of high activity Photosystem II membrane particles from spinach for starting population and cycle inefficiency (miss factor) and to perform a global fit of a data set to generate decay-associated spectra (DAS). These are made available at the Zenodo repository (doi: 10.5281/zenodo.7682034). Dependency versions: Python 3.7.9, Numpy 1.19.2, Scipy 1.5.2, Pandas 1.2.2., Matplotlib 3.3.2. |

For manuscripts utilizing custom algorithms or software that are central to the research but not yet described in published literature, software must be made available to editors and reviewers. We strongly encourage code deposition in a community repository (e.g. GitHub). See the Nature Portfolio guidelines for submitting code & software for further information.

## Data

Policy information about availability of data

All manuscripts must include a data availability statement. This statement should provide the following information, where applicable:
- Accession codes, unique identifiers, or web links for publicly available datasets
- A description of any restrictions on data availability
- For clinical datasets or third party data, please ensure that the statement adheres to our policy

The complete step-scan data obtained for 230.000 sequences of 10 laser flashes applied to dark-adapted PSII is available in form of averaged time courses of the detector signal for all 334 mirror positions and 10 exciting laser flashes per mirror position (deposited at Zenodo, doi: 10.5281/zenodo.7681840). We furthermore provide the spectrum used for correction of the so-called heat artefact at the same location. Using this data and the also deposited Python libraries, all steps of the data analysis can be reproduced.

The coordinates of the 14 MEP structures are provided as a Supplementary Data file in standard PDB format.

# Field-specific reporting

Please select the one below that is the best fit for your research. If you are not sure, read the appropriate sections before making your selection.

☒ Life sciences ☐ Behavioural & social sciences ☐ Ecological, evolutionary & environmental sciences

For a reference copy of the document with all sections, see nature.com/documents/nr-reporting-summary-flat.pdf

# Life sciences study design

All studies must disclose on these points even when the disclosure is negative.

| | |
|---|---|
| Sample size | For signal averaging in the step-scan experiment, about 230,000 measurement cycles (230,000 sequence of 10 laser flashes) were performed and the recorded data was averaged. Further details are reported in the Methods section of the article. The number of flash sequences was chosen such that the noise in the IR time courses could not affect our conclusions.<br>In panel e of the Extended Data Fig. 5, at each temperature the data points (filled circles) indicate the average of three values of TAUox obtained by simulation of the O2-transients of three independent experiments with averaging of 70-80 flash-induced O2-transients per experiment; the error bars indicate the corresponding standard deviation (n = 3). |
| Data exclusions | Prior to measurement, the OD of each spot is sampled, and those not meeting the criteria of 1 +/- 0.2 OD are excluded from further measurement. This typically resulted in the exclusion of about 10% of the spots. |
| Replication | We tracked 230,000 excitation cycles of dark-adapted photosystems with microsecond infrared spectroscopy. The resulting data was averaged in order to obtain a single high-quality data set. Further replicates were not approached. The data quality can be judged by visual inspection of time courses at specific wavenumbers shown in various figures of the article.<br>In panel e of the Extended Data Fig. 5, at each temperature the data points (filled circles) indicate the average of three values of TAUox obtained by simulation of the O2-transients of three independent experiments with averaging of 70-80 flash-induced O2-transients per experiment; the error bars indicate the corresponding standard deviation (n = 3). |
| Randomization | The topic of our study is a biological one, but the experimental biophysical approach and methodology is typical for physical sciences. It is not an empirical investigation where randomization can be applied. |
| Blinding | The topic of our study is a biological one, but the experimental biophysical approach and methodology is typical for physical sciences. It is not an empirical investigation where blinding can be applied. |

# Reporting for specific materials, systems and methods

We require information from authors about some types of materials, experimental systems and methods used in many studies. Here, indicate whether each material, system or method listed is relevant to your study. If you are not sure if a list item applies to your research, read the appropriate section before selecting a response.

## Materials & experimental systems

| n/a | Involved in the study |
|---|---|
| ☒ | ☐ Antibodies |
| ☒ | ☐ Eukaryotic cell lines |
| ☒ | ☐ Palaeontology and archaeology |
| ☒ | ☐ Animals and other organisms |
| ☒ | ☐ Human research participants |
| ☒ | ☐ Clinical data |
| ☒ | ☐ Dual use research of concern |

## Methods

| n/a | Involved in the study |
|---|---|
| ☒ | ☐ ChIP-seq |
| ☒ | ☐ Flow cytometry |
| ☒ | ☐ MRI-based neuroimaging |

