## [Peer Review File · Nature]

Manuscript Title: The electron-proton bottleneck of photosynthetic oxygen evolution

Reviewer Comments & Author Rebuttals

Reviewer Reports on the Initial Version:

Referees' comments:

Referee #1:

The new aspect of this manuscript is identification and characterization of the bottleneck of the reaction in PSII forming dioxygen from two water molecules. The bottleneck reaction step is identified as an electron transfer triggered concerted three-proton transfer process starting from protonated O6 via a water attached to Ca and further two waters attached to Mn4. The concerted proton movement lowers the enthalpy of the transition state, but at the same time, it lowers also the entropy of the transition state, giving rise to a large value of free energy in the transition state. This is so since at room temperature the thermal motion of the participating protons prevents coherence of proton movement. Hence, the necessary concerted proton movement to pass the reaction barrier occurs at random and therefore only rarely. The suggested reaction path of the concerted proton movement is verified by the agreement between measured and computed enthalpy (I could not find any details about the computed enthalpy). The true origin of the reaction bottleneck is the low entropy of the concerted proton movement. I would have expected a clearer discussion about this issue.

I would suggest to concentrate on data supporting the nature of the bottleneck in a letter version of the manuscript and provide more details in a separate publication elsewhere.

Some details below (text by the authors is in quotation):

The manuscript contains a large body of time-resolved infrared spectra of impressive detail. However, in my view all these details are not helping to understand the main subject of the manuscript.

"we exploited that the time constants of proton removal ($t_{H^+} = 340 \mu s$)"
Indicate which proton is removed and where did it go.

"Multiexponential simulations of the time courses provided 5 time-constants describing acceptor and donor side PSII processes, . . . "
Give the values of all five time constants and explain the underlying processes connected with them.

"The MEP calculations result in an energetic barrier of MnIV-O• formation in the S4->S4' transition of only 7 kcal/mol (300 meV)."
Unfortunately, I could not find any detail about this result of MEP computation neither in main text nor in the SI.

". . . the Glu65-Glu212 pair in Fig. 1c, which may described as a proton gate¹⁹ or proton loading site.²⁰ In analogy to the proton loading site of cytochrome c-oxidase,²¹"
It should be Glu312. This mistake occurs several times in the text. Stuchebruckhov first introduced the notion of proton loading site for cytochrome c oxidase. The authors should give correct credit.

"In an alternative scenario, we identify a proton-carrying carboxylate 'dyad', Glu312 and Glu65, as the site of deprotonation within about 300 μs after the flash, where presumably Glu312 is initially

protonated (ref.33, section SII.2 of the Supplementary 5 Information) and transiently deprotonated during the oxygen evolution transition. Other computational work suggests Glu65 as initially protonated,³⁴ but this difference here is of minor importance only."

Which one of the glutamates is protonated in the present work? The authors should make a clear statement also in the main manuscript and not just in the SI.

"As starting structures for the MEP, we used a linear interpolation between the initial and the final geometries of the selected reaction pathway."

What is the selected reaction pathway? There seems to be no information on it.

Some confusion, problems and suggestions with the figures:

It would be good to highlight the oxygen atoms that will form dioxygen. This should be done for Figs. 1c and 3b,c.

Caption of Fig. 1c: Ca is not green but pink.

Fig. 1: Indicate the values of the distances marked by dashed lines in the Fig. 1c.

The crystal structure reference should be cited for Fig. 1c.

Caption of Fig. 2:

"Red areas in b and d mark inverted 340 μ s DAS and 2.5 ms DAS" Red areas are in b and c and not in d.

"purple shaded areas in c mark similarity of 2.5 ms DAS" There is nowhere a shaded area and the purple area in in d and not c.

Fig. 2e is somewhat trivial and should be omitted.

The purpose Fig. 2f is not clear.

Fig. 3:

"creation of proton vacancy in the S3 \rightarrow S4": Indicate in panel b where the proton vacancy is localized.

Referee #2:

This paper addresses the mechanism of one of the most important biochemical processes on earth – the oxidation of water to O₂ in the oxygen evolving complex (OEC) of photosystem II (PSII). This reaction is the source of >90% of the earth's O₂ and has allowed the evolution of oxygenic organisms (like people) with much higher metabolic energy demands. The OEC is a remarkable catalyst as it carries out the difficult reduction of water at room temperature, in water at physiological pH with earth abundant metals. The OEC becomes successively oxidized in the Kok cycle so that the last oxidation can remove four electrons from substrate waters producing O₂ in the fully oxidized S₄ state. The states that lead to the reactive intermediates, S₀, S₁, S₂ and S₃, are metastable intermediates and have been characterized extensively, including by Dau. The transition from S₃ through the unstable S₄ to S₀ remains an area of real interest as this is where the chemistry gets done. As one cannot trap S₄, a melding of experiments and high-level simulations, as presented here, is required.

The paper first reports a heroic set of experiments that provides a remarkable set of time-resolved IR spectra. The data quality looks great. The main conclusion is that there is a transient deprotonation of a carboxylic acid group in the early phase of the S₃ to S₄ transition, that the loss of a proton is required to allow the final fourth oxidation of the OEC, which is required to generate O₂. This is a very important insight into a very important reaction. The second part of the paper is

a QM/MM analysis of suggested intermediates. This is a good solid analysis using standard techniques. The focus on the role of the protons as found experimentally. The simulations thus provide an important exploration of the consequences of the data.

This is an important piece of work (especially on the experimental side) and it certainly is appropriate for Nature. However, there are a number of questions that I have about the manuscript.

The key finding is that the time-resolved IR shows a loss of a proton from a residue when Yz is oxidized in the S3 state, and a regain of the proton prior to formation of S0. A protonation or deprotonation event led to a gain of peak in one part of the spectra and a loss in another. If the proton is transferred, two peaks should increase their amplitude and two lose amplitude. This should be discussed as evidence for a protonation/deprotonation event rather than a vibrational Stark shift.

The evidence for assigning the spectral changes at 1707, 1723, and 1744 to Glu 65 and 312 needs to be provided.

A key finding is that the amplitude before the third flash and again after formation of S0 are the same as a proton is only transiently lost. Kinetic traces that are provided do not support (or refute) the transient nature of the absorbance change because the x-axis does not have any pre-flash lead-in to the trace, so zero is not clear. Some pre-flash, 'baseline' data is needed to prove the point.

The authors realize there is a mixture of states at each timepoint. The parameters used to determine the mixture and the percentages of each at the third flash need to be given.

Glu 212 should be Glu 312.

The simulations are an important contribution to the paper exploring a mechanism based on the data. However, they are not the first to consider the nature of the transition from S3 to S0. These simulations need to be placed in context so a reader knows what is established/previously proposed (and by whom) to let us see what is new here.

The simulation should show the energy of the transition in the absence of the proposed proton shift. An implied conclusion is that the energy of OEC oxidation is lowered by proton loss. The paper would be strengthened with the calculation, or at least discussion, of the dependence of the OEC oxidation potential on the location and orientation of the protons directly.

The dependence of the energies on the orientation of the hydrogen bond network should be considered. The optimal hydrogen bond orientation may be very dependent on the number of waters and the choice of which parts of the proteins to include. The search for the optimized proton network should be described in methods, and the consequences for missing a better conformation should be noted.

Fig. 3b seems to imply the loss of a proton from Lys317, which does not seem correct.

The simulations note a higher-energy peroxide path, but I do not see a picture of the states in this alternative cycle or a mention of this mechanism in the text.

Minor points

The abstract could state the question more clearly. It seems to this reviewer that this paper highlights the role of proton transfers in the OEC rate determining step.

The data shows that the activation entropy is about half the activation energy, while the abstract overstated its importance.

SI Fig. 8 is valuable as it makes the mechanism clear. But the reader is left to puzzle over the exact nature of each change. It would be helpful if the legends provided a clear description of what is going on. Also, there are more dots on the graphs in Fig. 3 and SI Fig. 7 than intermediates shown in SI Fig. 8. The states in SI Fig. 8 should be matched to the locations on the energy diagram.

Another advantage of these pictures of the intermediates is it can reduce confusion about what to call states. For example, one can have a semantic argument about whether S4 starts when Yz or the OEC is oxidized. It would help if the text referred to these well-defined states in SI Fig. 8 when possible.

Fig. 1: The Kok cycle figures from the Dau lab are valuable, often copied into graduate student talks, so it's worth making them clear. The contribution of Yz/P680 on each step should be made clearer by at least mentioning it in the legend that this, while shown for only S1 to S2, occurs on every step.

Is the position of the H and E closer or further from the transition arrow meant to indicate PT or ET first? If so state this in the legend and refer to the earlier work establishing this.

Fig. 1, middle: It should be noted the rate constants are provided in the SI. The assignment of the band at 1384 cm⁻¹ should be given. Even if there is not an exact assignment of a particular residue it is likely reporting a kind of groups.

Fig. 2: Spectral traces should have an arrow showing the point where kinetic traces are taken from. Grey and green are hard to distinguish – make one bolder or dashed.

Fig. 3a: The nature of the + attached to the O or OH in the shaded region is unclear. Is this the OEC? Should another + be added in the middle frame when YZ is reduced? The identification of the lines in Fig. 3 is confusing. On the top it's not clear which lines refer to the right and to the left axis. The solid lines are in a different chart; however, while reading the legend I first looked for them in the top panel.

SI Fig. 6: The position of Yz should be indicated on the figure.

The recent paper by Yang et al. (Am. Chem. Soc. 2021. 143:8324-8332. doi: 10.1021/jacs.1c00633) concerning the proton trapping between Water1 and Asp 61 should be referenced.

Referee #3:

A. Summary of the key results

The authors present their results on the enigmatic photosynthesis process, for which Kok developed the mechanistic cycle with 5 S-state intermediates (S0, S1, S2, S3, S4). More specifically, the complexity of what happens between S4 and S0 is addressed and elucidated, i.e. the generation of O₂ and regeneration of the active site, the longest and thus rate-limiting step in photosynthesis. The authors address the interrelation between electron and proton transfer processes in these bottleneck steps of O₂ formation using time-resolved Fourier-transform infrared (FTIR) experiments, i.e. the oxygen evolution transition S₃ → S₄ → S₀ + O₂, and use computational techniques to present an insight into this reaction sequence.

Using DFT-based minimum energy path calculations, the authors study the S4 to S4' transition in detail (QMMM, NEB). The mechanism postulated for S4 to S4' includes hydrogen shuffling reactions with the aid of a solvent H2O molecule coordinated to Ca. In this process, the CaMn2O5(MnOH)2(H2O) active site rearranges to CaMn2O5(Mn-O*)(Mn-OH2)(H2O) and the formed water on Mn4 site jumps onto the carboxylic group of Asp61.

B. Originality and significance: if not novel, please include references

A key point herein is the creation of an M-O* radical, whose formation is estimated to be activated ~600 meV. This forms state S4' and from which O—O bond formation can happen through a binuclear mechanism between adjacent Mn sites. This can be seen as the direct coupling between a terminal Mn-oxyl radical with an oxo bridge between Ca and Mn, and was suggested in earlier works (based on a Nature Communications paper of Messinger et al. from 2014. DOI: 10.1038/ncomms5305):

Siegbahn, P. E. M. Water oxidation mechanism in photosystem II, including oxidations, proton release pathways, O-O bond formation and O2 release. *Biochim. Biophys. Acta* 1827, 1003–1019 (2013).

Cox, N. & Messinger, J. Reflections on substrate water and dioxygen formation. *Biochim. Biophys. Acta* 1827, 1020–1030 (2013).

Messinger, J. Evaluation of different mechanistic proposals for water oxidation in photosynthesis on the basis of Mn4OxCa structures for the catalytic site and spectroscopic data. *Phys. Chem. Chem. Phys.* 6, 4764–4771 (2004).

Vinyard, D. J., Ananyev, G. M. & Dismukes, G. C. Photosystem II: the reaction center of oxygenic photosynthesis. *Annu. Rev. Biochem.* 82, 577–606 (2013).

Siegbahn, P. E. M. O-O bond formation in the S4 state of the oxygen-evolving complex in photosystem II. *Chem. Eur. J.* 12, 9217–9227 (2006).

The whole photosynthesis pathway was first calculated by Siegbahn using DFT-techniques in the papers given above. Therefore, it is clear the computational work presented here builds on these earlier works and forms an elaboration on them using a larger model system (650,000 atoms) and molecular dynamics simulations and nudged elastic band simulations at the QM/MM level. It is, however, significant and impressive that very extensive NVT simulations (over several ns) were performed to estimate different geometric effects on states S3 and S4', however, with computational setups that were similar to the paper cited as ref. 68 (D. Narzi et al.). It would be good for the novelty if a biased MD approach could be used to study the S4 -> S4' transformation.

Furthermore, the video material provided of the pathway (based on the NEB-simulations) is insightful. Although it is clear that there are several novel elements in this research from the abstract, I would suggest that the authors emphasize the novelty of their work more specifically in the main text.

C. Data & methodology: validity of approach, quality of data, quality of presentation

The followed approach and combination between experimental and computational methods is suitable to elucidate the mechanism of photosynthesis.

For the quality of representations, some Ext. Data figures can be revised so their format and labelling is similar to the figures in the main text.

Related to the computational part, the following relatively remarks can be made:

(1) How does the electronic configuration of the active site change during the S4 → S4' conversion? This has not been discussed in detail (section NEB simulations with CP2K). The authors say they focus the high oxidation state paradigm; however, they cannot exclude the low oxidation state paradigm route. It is unclear how much both approaches differ in terms of energy. This should be calculated and discussed, and will allow the authors to make stronger statements related to this matter. Presently, it is unclear whether spin transitions are plausible during this S4 → S4' transition, and if so, this can point towards the necessity for a multi-reference approach to describe the active site.

(2) From the text, the entropic contribution to the free energy barrier for the S4 → S4' transition seems to be determined based on experimental data (Eyring equation). Can the entropy contribution not be calculated computationally as an additional verification of the suggested mechanism? What about the O2 elimination from S4' and regeneration of the active site with water molecules; what is the time required for these processes and do these not also contribute?

(3) The transfer from the Tyrz* hole to O6 in the S4' state can be computationally studied using a constrained DFT (CDFT) approach in CP2K to computationally verify the barrier connected to this charge transfer. The barrier for this transformation is assumed to be fast in the manuscript; however, this might be an underestimation.

(4) The energetics of several starting configurations (section classical MD simulations) are not discussed relative to each other.

(5) RMSD figures of the MD-simulations can be provided to check the convergence of the MD simulations.

(6) Units are used through each other (kcal/mol, meV).

(7) Can an agreement been found between the IR-spectra (Ext. Data Fig. 4), and calculated IR-spectra based on the studied model systems?

D. Appropriate use of statistics and treatment of uncertainties

There is appropriate use of statistics and treatment of uncertainties. For Ext. Data Fig. 5e; at some temperatures, there seem to be higher standard deviations on some of the measured time constants. The standard deviation of the time constants varies a lot for different temperatures. Is there any explanation for these variations?

E. Conclusions: robustness, validity, reliability

Conclusions are sound and suitable.

F. Suggested improvements: experiments, data for possible revision

Additional simulations and explanation of the computational part of the work (see under point C). Typos should be corrected in main text and supporting information.

G. References: appropriate credit to previous work?

The references appropriately credit previous work.

H. Clarity and context: lucidity of abstract/summary, appropriateness of abstract, introduction and conclusions

The clarity and context are sufficient. However, some typos should be corrected, and certain sentences can be rewritten for extra clarity.

Referee #4:

The manuscript presents a spectacular time-resolved IR spectroscopic study founded on a setup that allowed performing FTIR step-scan experiments on PSII with data collection spanning very long times. The focus is on the final steps that reset the cycle from the S3 to the S0 state while forming and evolving dioxygen. Analysis of the data resolves the kinetics of the transition(s); a major point is the identification of a specific time constant with an "obligatory" step of proton removal. The manuscript contains a solid computational part that supports and extends the experimental work toward plausible mechanistic details of the transformations that may occur leading up to O-O bond formation. The work is exceptional in all aspects and contains a wealth of information as well as careful and thoughtful discussion. It is a significant contribution to the field and I have no essential comments regarding either the experimental or the computational components of the study.

My main problem with the present work is the attribution of the label "S4" to the state associated with the obligatory proton removal. There is a fundamental difference in electronic structure depending on whether the hole is localized on the Mn4Ox cluster or on the tyrosine (Yz) that interfaces the cluster with P680. A metalloradical state of the OEC, i.e. a state with an oxidized tyrosyl radical (Yz*), is profoundly distinct in terms of its electronic and spectroscopic properties from a state with a reduced closed-shell Yz. These intermediates are clearly distinguished in approaches that are actually sensitive to where the electrons are located. For quite some time the S-state nomenclature is being used in (a large part of) the literature in a sense that reflects the electronic structure as understood in the above context (and not, for example, purely in the sense of kinetics). Thus, S0 and S0Yz* and S1 and S1Yz* and S2 and S2Yz* , and S3 and S3Yz* would all be distinct entities, both in theory and in practice (different properties, different observables – "pure" S-state versus "split signals" in EPR, X-ray spectroscopy etc). SiYz* versus S(i+1) simply means that in S(i+1) the hole has shifted from the tyrosyl radical to the Mn4Ox cluster – the OEC has transitioned. However, what the manuscript labels as "S4" is merely a form of S3Yz*. The S3Yz* intermediate is a classical metalloradical intermediate. This has been likely observed already by magnetic resonance spectroscopy, see e.g. FEBS Lett. 2014, 588:1827, and Photosynth. Res. 2016, 130:417. Is the EPR-observed intermediate the same as the here reported intermediate? This is an important question in itself that the authors must consider and comment upon. But whether this is the same or not, it is hard to justify calling it "S4" when the hole is still on Yz*. This would mean applying different nomenclature and different criteria for different use cases or for different experimental techniques. If we apply the same logic and the same criteria here as in all other transitions, then the only species discussed in this work that is really "S4" is the intermediate where an oxyl radical is formed on the cluster. This is also the same way such intermediates are discussed in the totality of the theoretical literature I am aware of. A deprotonation or any other event can of course occur while the system is poised electronically in a metalloradical Yz* state. This is well known for S2Yz*, where a lot of things do happen and several variants, including deprotonated forms, have been observed and characterized (see for example Boussac's work on these intermediates). None of the various types of S2Yz* intermediates, deprotonated or not, would ever be called "S3". Thus, using "S4" for a subsequent S3Yz* metalloradical intermediate, regardless of the protonation events and the kinetics involved, feels

wrong to me.

Secondary points:

- Use chain identifiers when aminoacids are referred to, at least when they are mentioned for the first time. For example, is the Glu65–Glu212 pair in the D1 protein?
- Page 3 mentions the Glu65-Glu212 pair, but Fig. 3 shows Glu65-Glu312.
- On p. 5 the phrase “Noteworthy, Yano and coworkers...” is accompanied by Ref. 21, but this is not the correct reference.
- Page 5, “... which is in line with a mechanistic proposal [...] and recent structural data.”: Here Refs. 22 and 37 are cited, but the statement is not really supported by the references. Kern et al. is not a study that can support any correspondence between the published models and the present mechanism, while Ref. 37 is in fact inconsistent with the mechanism and the nature of intermediates proposed here. Suga et al. report a structural model (and accompanying calculations) that assign the presence of an oxyl radical in the S3 state (i.e. they conclude that there is no Mn-centered oxidation in the S2-S3 transition). This is not at all what the authors of the present manuscript propose.
- Also on p. 5, “We emphasize that Siegbahn and others...”: Here Ref. 40 is obviously a mistake; it is a study of pK values that is irrelevant here. It can be replaced by one of many studies on the mechanism by Siegbahn or others. I am not sure that the statement itself is entirely correct; one should have a closer look into some of the papers by Soji et al. (e.g. some 2018-2019 Chem. Phys. Letters).

Author Rebuttals to Initial Comments:

We thank the four reviewers for their positive, detailed and very constructive comments. In an effort to address these comprehensively, we have performed additional analyses and also complemented the Supplementary Information with additional experimental data. The additional data and their analysis also reflect discussions at the International Photosynthesis Congress in New Zealand and its satellite meeting on photosynthetic oxygen evolution, where the work was well received. We apologize for the delay, but believe it was worth investing the time in striving for clear and unambiguous results.

Referee #1:

1-1) The new aspect of this manuscript is identification and characterization of the bottleneck of the reaction in PSII forming dioxygen from two water molecules. The bottleneck reaction step is identified as an electron transfer triggered concerted three-proton transfer process starting from protonated O6 via a water attached to Ca and further two waters attached to Mn4. The concerted proton movement lowers the enthalpy of the transition state, but at the same time, it lowers also the entropy of the transition state, giving rise to a large value of free energy in the transition state. This is so since at room temperature the thermal motion of the participating protons prevents coherence of proton movement. Hence, the necessary concerted proton movement to pass the reaction barrier occurs at random and therefore only rarely. The suggested reaction path of the concerted proton movement is verified by the agreement between measured and computed enthalpy (I could not find any details about the computed enthalpy). The true origin of the reaction bottleneck is the low entropy of the concerted proton movement. I would have expected a clearer discussion about this issue.

We thank the reviewer for this concise and precise assessment of our central conclusions. The reviewer asks (i) for more information on the computed enthalpy and (ii) a clearer discussion of the entropic contribution of the free-energy of activation associated with the concerted proton movement. Both requests overlap with other reviewer questions and are extensively dealt with in the revised manuscript. The method for calculation of the reaction path is detailed in the Methods section of the article as section "Minimum Energy Path calculations" and extensively dealt with in Supplementary Information SII.4; all structures of the reaction-path intermediates are now shown in Suppl. Fig. S14 and the coordinates are provided as a separate Ascii file. The entropic contribution is addressed in the main text and more extensively in two new sections of the Supplementary Information (SII.8 and SII.9).

1-2) I would suggest to concentrate on data supporting the nature of the bottleneck in a letter version of the manuscript and provide more details in a separate publication elsewhere. Some details below (text by the authors is in quotation):

The manuscript contains a large body of time-resolved infrared spectra of impressive detail. However, in my view all these details are not helping to understand the main subject of the manuscript.

The presented experimental data as well as the computational analysis, both are focused on the bottleneck step in the $S_3 \rightarrow S_0$ transition. In the revised manuscript, this focus has been strengthened further, inter alia by additional results provided in the SI that address exclusively the $S_3 \rightarrow S_0$ transition; a separate publication on other S-state transitions is in preparation, as suggested by the reviewer. Yet we consider it as important to show (i) exemplary data also for other S-state transitions and (ii) various spectra of intermediate stages of the data analysis, the latter in form of Supplementary Information. Thereby, (i) also the non-expert readers can understand the "spirit" of the experimental approach (Figure 1b, Extended Data Figures 1 and 2) and (ii) the detailed information provided as Supplementary Information ensures that if approached, other researchers could reproduce experiments and data evaluation.

1-3) "we exploited that the time constants of proton removal ($t_{H^+} = 340 \mu s$)" Indicate which proton is removed and where did it go.

We now have indicated by arrows and circles the binding sites of the functional relevant protons and their movements in several figures, inter alia in Figure 2 and Extended Data Fig. 8.

1-4) "Multiexponential simulations of the time courses provided 5 time-constants describing acceptor and donor side PSII processes, . . ." Give the values of all five time constants and explain the underlying processes connected with them.

The presentation of the time constant values may have been too hidden in the previous SI version. We now point more explicitly to the respective SI tables providing the time constants and their assignment (Suppl. Table 2 and Suppl. Table 3).

1-5) "The MEP calculations result in an energetic barrier of MnIV-O● formation in the S4->S4' transition of only 7 kcal/mol (300 meV)."

Unfortunately, I could not find any detail about this result of MEP computation neither in main text nor in the SI.

In the revised manuscript, the method for calculation of the reaction path is detailed in the Methods section of the article as section "Minimum Energy Path calculations". Further information is provided in Supplementary Information SII.4, including the employed quantum region (Suppl Fig. 18) and a scheme summarizing the here used nudged-band method (Suppl. Fig. 19). All structures of the reaction-path intermediates are shown (Suppl. Fig. S20) and their coordinates are provided in form of Ascii file for download (in PDB format).

1-6) ". . . the Glu65-Glu212 pair in Fig. 1c, which may described as a proton gate¹⁹ or proton loading site.²⁰ In analogy to the proton loading site of cytochrome c-oxidase,²¹" It should be Glu312. This mistake occurs several times in the text. Stuchebruckhov first introduced the notion of proton loading site for cytochrome c oxidase. The authors should give correct credit.

We thank the reviewer for pointing out the error in numbering of a crucial residue, which has been corrected. We also thank for mentioning the work of Stuchebruckhov, which now is cited.

1-7) "In an alternative scenario, we identify a proton-carrying carboxylate 'dyad', Glu312 and Glu65, as the site of deprotonation within about 300 μs after the flash, where presumably Glu312 is initially protonated (ref.³³, section SII.2 of the Supplementary 5 Information) and transiently deprotonated during the oxygen evolution transition. Other computational work suggests Glu65 as initially protonated,³⁴ but this difference here is of minor importance only."

Which one of the glutamates is protonated in the present work? The authors should make a clear statement also in the main manuscript and not just in the SI.

Following the reviewer's request, the above sentence has been rephrased: "In an alternative scenario, we identify a proton-carrying carboxylate 'dyad', Glu312 and Glu65, as the site of deprotonation within about 300 μs after the flash. *Here we assume that Glu312 is initially protonated* (ref.³⁷, section SII.2 of the Supplementary Information) and transiently deprotonated during the oxygen evolution transition. Other computational work suggests Glu65 as initially protonated,³⁸ but this difference here is of minor importance only."

1-8) "As starting structures for the MEP, we used a linear interpolation between the initial and the final geometries of the selected reaction pathway."

What is the selected reaction pathway? There seems to be no information on it.

See our response to 1-5), which is provided further above.

1-9) Some confusion, problems and suggestions with the figures:

We thank the reviewer for pointing out errors and suggesting valuable improvements in the article figures. We have addressed all these suggestions as detailed in the following.

a) It would be good to highlight the oxygen atoms that will form dioxygen. This should be done for Figs. 1c and 3b,c.

The oxygen atoms that form dioxygen are now highlighted in Fig. 1c, Fig. 3b and Fig. 3c, as suggested by the reviewer.

b) Caption of Fig. 1c: Ca is not green but pink.

Thank you for pointing out this error; the figure caption has been corrected accordingly.

c) Fig. 1: Indicate the values of the distances marked by dashed lines in the Fig. 1c.

We also feel that the H-bond distances represent valuable information. For maintaining clarity in Figure 1c, the H-bonding distances as well as the assignment of the shown residues to polypeptide chains are now indicated in an additional Suppl. Information Fig. 1; in the figure caption of Figure 1c we refer to this SI figure.

d) The crystal structure reference should be cited for Fig. 1c.

The reference is cited in the figure caption.

e) Caption of Fig. 2: “Red areas in b and d mark inverted 340 μ s DAS and 2.5 ms DAS” Red areas are in b and c and not in d. // “purple shaded areas in c mark similarity of 2.5 ms DAS” There is nowhere a shaded area and the purple area in in d and not c,

The error in the figure caption has been corrected.

f) Fig. 2e is somewhat trivial and should be omitted. The purpose Fig. 2f is not clear.

In response to the reviewer’s comment, the illustrative panels e and f have been removed.

g) Fig. 3: “creation of proton vacancy in the S3→ S4”: Indicate in panel b where the proton vacancy is localized.

We have followed the reviewer's suggestions (a blue dotted-line circle around the proton in Fig. 3b and empty circle in 3c).

Referee #2:

This paper addresses the mechanism of one of the most important biochemical processes on earth – the oxidation of water to O₂ in the oxygen evolving complex (OEC) of photosystem II (PSII). This reaction is the source of >90% of the earth's O₂ and has allowed the evolution of oxygenic organisms (like people) with much higher metabolic energy demands. The OEC is a remarkable catalyst as it carries out the difficult reduction of water at room temperature, in water at physiological pH with earth abundant metals. The OEC becomes successively oxidized in the Kok cycle so that the last oxidation can remove four electrons from substrate waters producing O₂ in the fully oxidized S₄ state. The states that lead to the reactive intermediates, S₀, S₁, S₂ and S₃, are metastable intermediates and have been characterized extensively, including by Dau. The transition from S₃ through the unstable S₄ to S₀ remains an area of real interest as this is where the chemistry gets done. As one cannot trap S₄, a melding of experiments and high-level simulations, as presented here, is required.

The paper first reports a heroic set of experiments that provides a remarkable set of time-resolved IR spectra. The data quality looks great. The main conclusion is that there is a transient deprotonation of a carboxylic acid group in the early phase of the S₃ to S₄ transition, that the loss of a proton is required to allow the final fourth oxidation of the OEC, which is required to generate O₂. This is a very important insight into a very important reaction. The second part of the paper is a QM/MM analysis of suggested intermediates. This is a good solid analysis using standard techniques. The focus on the role of the protons as found experimentally. The simulations thus provide an important exploration of the consequences of the data.

This is an important piece of work (especially on the experimental side) and it certainly is appropriate for Nature. However, there are a number of questions that I have about the manuscript.

We thank the reviewer for positive assessment and constructive suggestions, which are addressed in the following.

2-1) The key finding is that the time-resolved IR shows a loss of a proton from a residue when Yz is oxidized in the S₃ state, and a regain of the proton prior to formation of S₀. A protonation or deprotonation event led to a gain of peak in one part of the spectra and a loss in another. If the proton is transferred, two peaks should increase their amplitude and two lose amplitude. This should be discussed as evidence for a protonation/deprotonation event rather than a vibrational Stark shift.

This is a highly important comment addressing a key point of our results. Therefore, in response to both this reviewer comment and constructive discussions we had at the International Photosynthesis Congress and its satellite meeting on Oxygen Evolution (New Zealand, July/August 2022), we now complement our previously provided data with additional experimental results and analyses.

Our time-resolved IR data indicates carboxylate deprotonation; the data is not explainable by a vibrational Stark shift effect. *To evidence and clarify this point, in the Suppl. Information of the revised manuscript we now provide significant additional data and analyses* addressing (i) the potential overlap with Stark-shift bands related to Tyr_z oxidation and reduction (Suppl. Information SI.9 and SI.10) (ii) an additional step-scan data set with H₂O exchanged against D₂O (Suppl. Information SI.10). The evidence for transient deprotonation of a carboxylate sidechains and specifically of the E65-E312 pair is summarized in Suppl. Information SI.11.

2-2) The evidence for assigning the spectral changes at 1707, 1723 and 1744 to Glu 65 and 312 needs to be provided.

In line with the reviewer, we consider the evidence for assignment of these bands to the Glu65-Glu312 dyad a point of special importance. Therefore, in addition to the arguments provided in the main article, this evidence is now summarized in a special section of the Supporting Information (SI.11).

2-3) A key finding is that the amplitude before the third flash and again after formation of S0 are the same as a proton is only transiently lost. Kinetic traces that are provided do not support (or refute) the transient nature of the absorbance change because the x-axis does not have any pre-flash lead-in to the trace, so zero is not clear. Some pre-flash, 'baseline' data is needed to prove the point.

Following the reviewer's suggestion, the pre-flash data now is shown in Figure 2b and Extended Data Figure 4.

2-4) The authors realize there is a mixture of states at each timepoint. The parameters used to determine the mixture and the percentages of each at the third flash need to be given.

A table with the S-state populations after each flash now is shown as Suppl. Table 1, along with the deconvolution parameters. We emphasize that the deconvolution (described in Suppl. Information SI.5) does not only rely on the data sets obtained by the Flash 1 to Flash 4 (or Flash 2 to Flash 5) but considers all data sets from Flash 2 to Flash 10.

2-5) Glu 212 should be Glu 312.

We apologize for this error, which has been corrected throughout main manuscript and SI.

2-6) The simulations are an important contribution to the paper exploring a mechanism based on the data. However, they are not the first to consider the nature of the transition from S3 to S0. These simulations need to be placed in context so a reader knows what is established/previously proposed (and by whom) to let us see what is new here.

We consider this an especially important comment, which overlaps with a suggestion of Reviewer 3. In the revised manuscript, our results are placed into the context of established and previously proposed mechanistic proposals and computational results clearly more strongly, now including overall 18 related citations in the main article text.

In the revised manuscript, the novel and distinguishing aspects of our computational analyses are explicitly clarified in the main article text on page 4 (top paragraph + last sentence of second paragraph) and in the first paragraph of the Conclusion section (bottom of page 5, top of page 6). Moreover, because utmost brevity is required in the main article, we have added a section on this point as Suppl. Information SI.6.

We believe that our study generally is distinguished from all previous ones by establishing a close relation between the sequence of events and kinetic data concluded from experimental findings, on the one hand, and the computational analyses, on the other hand. This combination allows the identification of both the essential priming step of proton release from the Glu65-Glu312 dyad and the rate-limiting step of photosynthetic O₂ formation. Enabled by a priming step of proton release, the rate-limiting step is a specific mode of oxyl-radical formation by coupled electron and multi-proton transfer, which could be identified only by a computational approach that has not been realized before. It is a single minimum energy-path calculation that describes not only O-O bond formation, but also the decisive preceding events leading to oxyl-radical formation by electron transfer from the metal cluster coupled to proton movements in the vicinity of the oxidant, Y₂^{ox}, and

the metal cluster itself. For more details, see main text sections listed above (marked by yellow background in the article version with marked changes) and Suppl. Information SII.6.

2-7) The simulation should show the energy of the transition in the absence of the proposed proton shift. An implied conclusion is that the energy of OEC oxidation is lowered by proton loss. The paper would be strengthened with the calculation, or at least discussion, of the dependence of the OEC oxidation potential on the location and orientation of the protons directly.

Following the suggestion of the reviewer, we have carried out further calculations. We considered two cases: the first, in which the protons move before the electron transfer; the second, the electron transfer from O6 occurs without protons movements. In the first case we constrained the proton shared between TyrZ and His190 to bond the His nitrogen, therefore forcing the TyrZ in its radical form. In this situation, the energy variation associated with the simple proton transfer destabilizes the system by +7.5 kcal/mol with respect to the starting structure. In the second case (electron transfer without proton movements) we failed even to converge a single point electronic structure calculation, strongly suggesting that this latter hypothesis must be excluded. These computational results are now included in Suppl. Information in section SII.7.

2-8) The dependence of the energies on the orientation of the hydrogen bond network should be considered. The optimal hydrogen bond orientation may be very dependent on the number of waters and the choice of which parts of the proteins to include. The search for the optimized proton network should be described in methods, and the consequences for missing a better conformation should be noted.

The starting structures used in the MEP calculations were extracted from QM/MM MD simulations, therefore including the protein environment around the QM region and taking into account the thermal fluctuations. Extracting the initial and final structures of the MEP calculation from QM/MM MD simulations at finite temperature help to find an optimal hydrogen bond network for the water and the residues included in the simulated system. Still, we cannot exclude that an even more favorable hydrogen bond network could be reached, for example increasing the simulation time of the QM/MM MD trajectory. We pointed out these considerations in the Methods part, subsection "Minimum Energy Path calculations".

2-9) Fig. 3b seems to imply the loss of a proton from Lys317, which does not seem correct.

We have improved the graphical presentation in Figure 3B and elsewhere. Now blue arrows starting at a H⁺ position indicate the specific proton that is moving, clarifying that the proton from the Glu65-Glu312 dyad is lost. The movement of the Lys residue occurs without protonation state changes as now is indicated by a broad grey arrow.

2-10) The simulations note a higher-energy peroxide path, but I do not see a picture of the states in this alternative cycle or a mention of this mechanism in the text.

The energy barrier associated with the peroxide-bond formation is not higher than the energy barrier associated with the O6 radicalization coupled to Asp61 protonation. From an enthalpic point of view they are comparable (as shown by MEP calculations, see e.g. Fig. 3), while from an entropic point of view, our results indicate that the latter is increased by 6.5 kcal/mol. In contrast, we do not expect that the energy barrier for the peroxide-bond formation will be significantly affected by entropy. An improved explanation of the source of the entropic contribution now is presented in the revised version of the manuscript should avoid the misunderstanding (pages 4-5 in the main manuscript and SII.9 in supplementary information).

We also stated that oxyl radical formation without invoking proton-coupled electron transfer “would involve an energetically unfavorable intermediate state.” This assertion is now supported by calculations reported in Supp. Information SII.7.

Minor points

2-11) The abstract could state the question more clearly. It seems to this reviewer that this paper highlights the role of proton transfers in the OEC rate determining step.

The central questions we intend to address is the nature of both the overall rate-determining step of oxygenic photosynthesis and of the preceding (enabling) proton removal step. Both questions relate directly to the enigmatic nature of the S_4 -state, which has riddled photosynthesis researchers for more than 50 years and also has driven our research. Proton transfer is indeed crucial in our study, as emphasized twice in the abstract: “*crucial proton vacancy*”, “*astonishing single-electron multi-proton transfer event*”, and strongly emphasized in the title (“*The electron-proton bottleneck of photosynthetic oxygen evolution*”). All in all, we feel that these questions are well addressed in the abstract, within the limits set by the instructions of Nature for content and organization of the ‘introductory paragraph’.

2-12) The data shows that the activation entropy is about half the activation energy, while the abstract overstated its importance.

We see the point made by the reviewer and the abstract has been changed accordingly. We now say: “*Subsequently, a reactive oxygen radical is formed in an astonishing single-electron multi-proton transfer event. This is the slowest step in photosynthetic O₂-formation, with moderate energetic barrier and remarkable entropic slowdown.*”

2-13) SI Fig. 8 is valuable as it makes the mechanism clear. But the reader is left to puzzle over the exact nature of each change. It would be helpful if the legends provided a clear description of what is going on. Also, there are more dots on the graphs in Fig. 3 and SI Fig. 7 than intermediates shown in SI Fig. 8. The states in SI Fig. 8 should be matched to the locations on the energy diagram.

We assume that the reviewer is referring to Extended Data Fig. 8. *Its figure caption has been extended significantly and now clarifies how the structures of the shown intermediate states were obtained.* The dots in the graphs of Fig. 3 and Extended Data Fig. 7 do not relate directly to the structures of the intermediate states shown in Extended Data Fig. 8. They relate to reaction-path structures of the MEP calculations, which address the transition from the structures (3) via (4) to (5) of Extended Data Fig. 8. The reaction-path structures of the MEP calculations are shown in the Suppl. Information Fig. 20 and also provided also in form of a PDP file.

Another advantage of these pictures of the intermediates is it can reduce confusion about what to call states. For example, one can have a semantic argument about whether S_4 starts when Yz or the OEC is oxidized. It would help if the text referred to these well-defined states in SI Fig. 8 when possible.

We discussed this semantic, yet important issue of naming individual states during the International Photosynthesis Congress in New Zealand and its Oxygen Evolution satellite (July/August 2022). Following the consensus view expressed there, *we have changed the nomenclature in Figure 8 and consequently throughout the manuscript (S_4 to S_3' , S_4' to S_4), as also explained in the main text of the manuscript on page 6 (second paragraph).*

2-14) Fig. 1: The Kok cycle figures from the Dau lab are valuable, often copied into graduate student talks, so it's worth making them clear. The contribution of Yz/P680 on each step should be made clearer by at least mentioning it in the legend that this, while shown for only S1 to S2, occurs on every step.

We are pleased hearing that our Kok cycle figure are often copied by graduate students. In the revised manuscript, Figure 1a has been modified such that that the electron transfer to Tyr_z (and from Tyr_z to P680 occurs on each step. Further intermediates relating to proton transfer in the S₂->S₃ and S₀->S₁ are not shown to keep the scheme simple, because we focus on intermediates in the S₃->S₀ transition.

2-15) Is the position of the H and E closer or further from the transition arrow meant to indicate PT or ET first? If so state this in the legend and refer to the earlier work establishing this.

The sequence of electron and proton removal from the Mn₄Ca-oxo cluster is more clearly presented in the revised scheme of Figure 1a; a review article proposing this sequence and two original experimental studies establishing it are now cited in the figure caption.

2-16) Fig. 1, middle: It should be noted the rate constants are provided in the SI. The assignment of the band at 1384 cm⁻¹ should be given. Even if there is not an exact assignment of a particular residue it is likely reporting a kind of groups.

Following the suggestion of the reviewer, we now say in the figure caption: "At 1384 cm⁻¹, the IR transients predominantly reflect symmetric carboxylate-stretching vibrations of protein sidechains that are sensitive to Mn oxidation and reduction. The colored lines represent simulations, for which the time constant values (inverse rate constants) are provided in Supplementary Tab. 2."

2-17) Fig. 2: Spectral traces should have an arrow showing the point where kinetic traces are taken from. Grey and green are hard to distinguish – make one bolder or dashed.

Following the reviewer's suggestions, in the spectra of panel b, c, and d of Figure 2, the corresponding wavenumbers of the transients shown in panel-a are now clearly marked by colored asterisks (*). The grey line has been replaced by a broken black line.

2-18) Fig. 3a: The nature of the + attached to the O or OH in the shaded region is unclear. Is this the OEC? Should another + be added in the middle frame when YZ is reduced? The identification of the lines in Fig. 3 is confusing. On the top it's not clear which lines refer to the right and to the left axis. The solid lines are in a different chart, while reading the legend made me look for.

We have revised Figure 3a for improved clarity. Also, the corresponding figure caption has been extended and now provides more explanation of the scheme of panel-a.

2-20) SI Fig. 6: The position of Yz should be indicated on the figure.

The Extended Data Fig. 6 has been replaced with an improved version, where also the redox-active tyrosine is clearly labeled.

2-21) The recent paper by Yang et al. (Am. Chem. Soc. 2021. 143:8324-8332. doi: 10.1021/jacs.1c00633) concerning the proton trapping between Water 1 and Asp 61 should be referenced.

This study now is cited on page 6 of the main text.

Referee #3:

We thank the reviewer for positive assessment and instructive suggestions, which are thoroughly addressed in the following.

3-1) The whole photosynthesis pathway was first calculated by Siegbahn using DFT-techniques in the papers given above. Therefore, it is clear the computational work presented here builds on these earlier works and forms an elaboration on them using a larger model system (650,000 atoms) and molecular dynamics simulations and nudged elastic band simulations at the QM/MM level. It is, however, significant and impressive that very extensive NVT simulations (over several ns) were performed to estimate different geometric effects on states S3 and S4', however, with computational setups that were similar to the paper cited as ref. 68 (D. Narzi et al.). It would be good for the novelty if a biased MD approach could be used to study the S4 -> S4' transformation.

In the revised manuscript, the relation to earlier work as well as the novelty of our study are explicitly clarified in the main article text on page 4 (top paragraph + last sentence of second paragraph) and in the first paragraph of the Conclusion section (bottom of page 5, top of page 6). Moreover, because utmost brevity is required in the main article, we have added a section on this point as Suppl. Information SII.6. The rationale and used methodology for calculation of the reaction path is detailed in the Methods section of the article in a special subsection "Minimum Energy Path calculations" and extensively dealt with in Supplementary Information SII.4.

We agree with the referee that the present theoretical work is built on the shoulders of seminal previous calculations, in particular by Per Siegbahn and later others. These provided a starting point for the evaluation of complex calculations such as the present nudged elastic band simulations of the large catalytic complex. Such calculations are presently at the edge of the computational capabilities in term of accuracy and size. The referee's suggestion of using a biased MD approach to follow the oxyl-radical formation is potentially interesting, but we believe that it is currently not technically feasible, and probably it will not add substantial quantitative information regarding the reaction step. Indeed, we have shown that the entropic contribution of the reaction in our case is large, and it is likely due to complex rearrangements of first and second sphere H-bond networks. These rearrangements not easily caught by a collective variable and biased QM/MM techniques cannot provide a feasible solution to explore such movements. Such complex rearrangements would indeed require long integration times and large phase space exploration even when using biased QM/MM molecular dynamics approaches; such calculations are therefore currently beyond the limits of current computational capabilities. Although a further investigation can in principle provide more insights on the exact details of the hydrogen bond rearrangement which is responsible for entropic contribution, we believe that the calculated energy barrier would have a similar height with respect to the present minimum energy path calculations, therefore not altering the proposed mechanism.

3-2) Furthermore, the video material provided of the pathway (based on the NEB-simulations) is insightful. Although it is clear that there are several novel elements in this research from the abstract, I would suggest that the authors emphasize the novelty of their work more specifically in the main text.

Following the reviewer's suggestion, now we highlight more strongly the novelty of our work in the main article text on page 4 (top paragraph + last sentence of second paragraph) and in the first paragraph of the Conclusion section (bottom of page 5, top of page 6), and in a special section of the

Supplementary Information (SII.6). We also thank the reviewer for the appreciation of the multimedia material. The video is now complemented by an Ascii file providing the underlying structures along the transition path for the complete quantum region in PDB format.

We believe that our study generally is distinguished from previous ones by establishing a close relation between the sequence of events and reaction kinetics concluded from experimental findings, on the one hand, and the computational analyses, on the other hand. This combination enables identification of both, the essential priming step of proton release from the Glu65-Glu312 dyad and the rate-determining step of photosynthetic O₂-formation. The rate-limiting steps is a specific mode of oxyl-radical formation by coupled electron and multi-proton transfer, which could be identified only by a computational approach that has not been realized before. It is a single minimum energy-path calculation that describes not only O-O bond formation, but also the decisive preceding events leading to oxyl-radical formation by electron transfer from the metal cluster coupled to proton movements in the vicinity of the oxidant, Tyr_Z^{ox}, and the metal cluster itself. We emphasize a further novel aspect that previous investigations did not consider, namely the mechanistic features of a combined enthalpic—entropic barrier associated with the Mn-OH deprotonation and Mn^{IV}-O• radical formation, which we identify as the kinetically most demanding, overall bottleneck step in photosynthetic water oxidation.

Established and previously proposed calculations and mechanisms are discussed in the main manuscript (first paragraph of Conclusion section on pg. 5/6), but comparably briefly only, because utmost brevity is required in the main article. Therefore, we also have added a discussion section on this point in the Supporting Information, see SII.6.

3-3) For the quality of representations, some Ext. Data figures can be revised so their format and labelling is similar to the figures in the main text.

Following the suggestion of the reviewer, we have revised several figures of the main article and of the Extended Data Figures for more coherent general appearance.

Related to the computational part, the following relatively remarks can be made:

3-4) (1) How does the electronic configuration of the active site change during the S₄ -> S₄' conversion? This has not been discussed in detail (section NEB simulations with CP2K). The authors say they focus the high oxidation state paradigm; however, they cannot exclude the low oxidation state paradigm route. It is unclear how much both approaches differ in terms of energy. This should be calculated and discussed, and will allow the authors to make stronger statements related to this matter. Presently, it is unclear whether spin transitions are plausible during this S₄ -> S₄' transition, and if so, this can point towards the necessity for a multi-reference approach to describe the active site.

Calculations for on reactions of the S-state based on the low-oxidation state paradigm would seriously conflict with experimental findings and preclude a consistent computational approach for the complete S-state cycle, as detailed below. References are provided to clarify why our calculations are based on the high oxidation-state paradigm (on pg. 14, first paragraph of section on MEP calculations). The potentially misleading statement that the low-oxidation state paradigm had been included for fairness to an outsider point of view, but now has been dropped for clarity.

The electronic configuration of the active site can be followed by the lower panel of Figure 3a of the main text. It can be noted that the electron transfer occurs concurrently with the other proton transfers on D61 and the proton release from O6.

There is very strong experimental and theoretical evidence in favor of the high-oxidation paradigm which is supported by most of the computational and experimental evidence, as convincingly

summarized by Pantazis and coworkers [Krewald et al, 2015, Chem. Sci. 6, 1676-1695]. We note in passing that also Per Siegbahn in his seminal computational studies always has excluded the low-oxidation state scenario. All our calculations were consistent with the high-oxidation paradigm, starting from the beginning, i.e., the S1 state characterization [Narzi et al. (2017). Chemistry–A European Journal, 23, 6969-6973.], and following with the studies of the pathways through S2, S3 and S4 [Narzi et al. (2014). Proceedings of the National Academy of Sciences, 111, 8723-8728; Capone et al. (2020) Chemical Physics Letters, 742, 137111; Nakamura et al. (2020). Physical Chemistry Chemical Physics, 22, 273-285]. The structure and the spectroscopic fingerprints (EPR and IR for instance) of these theoretical models have been validated in the last decade against many experimental data, providing a solid and consistent pathway of the Kok's S-state cycle.

Aside from being against the majority of the experimental evidence, an accurate evaluation of the low-energy pathway can be in principle correctly considered only if firstly validated against all the experimental data for which the high-oxidation pathway has been validated. Part of this work has been already carried out by other groups [Cox et al. (2020). Annual review of Biochemistry, 89, 795-820.], confirming the strong preference for the high-oxidation pathway. *We might in principle calculate the energies of the two transition states and the 3 intermediate states reported in the present work using the structures taken from the low oxidation state paradigm, but this would offer an unreliable estimation of the barrier since the full catalytic pathway should be reconsidered, starting from S₀ or at least from new S₃ QMMM-MD simulations carried on in the high oxidation paradigm.*

3-5) (2) From the text, the entropic contribution to the free energy barrier for the S4 → S4' transition seems to be determined based on experimental data (Eyring equation). Can the entropy contribution not be calculated computationally as an additional verification of the suggested mechanism? What about the O₂ elimination from S4' and regeneration of the active site with water molecules; what is the time required for these processes and do these not also contribute?

In the revised version of the manuscript, we have added a section in the Supplementary Information (SII.9) which discusses computational approaches towards the entropic contribution (to the rate-determining step of the O₂-formation chemistry), inter alia illustrating its role by presenting results from ab-initio MD simulations in Suppl. Fig. 22.

A direct calculation of the entropic contribution is not feasible with the current computational capabilities due to the long simulation time required and due to the necessity to include second shell ligands. The configurational entropy of the hydrogen bond network cannot be indeed estimated by harmonic approximation, but it would require techniques such thermodynamic integration, which necessitates the use of large phase space sampling on quantum potential energy surfaces. We note that O₂ release and Mn₄Ca-oxo cluster restoration has been investigated in our recent work [Capone et al. Biochemistry 2021, 60, 2341–2348] and it is contributing accordingly to the total mechanism reported in Extended Data Fig. 8. In the present work, we are focused on the rate-determining step, which does not include the cluster restoration step.

3-6) (3) The transfer from the Tyrz* hole to O6 in the S4' state can be computationally studied using a constrained DFT (CDFT) approach in CP2K to computationally verify the barrier connected to this charge transfer. The barrier for this transformation is assumed to be fast in the manuscript; however, this might be an underestimation.

The relation of the used computational approach to the rate of electron transfer is now discussed in an additional section of the Supplementary Information (SII.8).

In our calculations we are not making assumptions on the time required for the electron transfer. The Constrained DFT approach can be a valuable tool to find “electronic reaction coordinates”

to explore the potential energy surface of reactions. In the present work we adopt a different approach by performing minimum energy path calculations. Our unique assumption is in the structure of the starting and ending points of the path, all the other intermediates are just a consequence of the nudged elastic band calculations. For an in-depth discussion on the relation of our calculations to electron transfer theory, we have added in Supplementary Information the section SII.8.

3-7) (4) The energetics of several starting configurations (section classical MD simulations) are not discussed relative to each other.

Our starting MD simulations are all generated by a single configuration which is based on the crystallography structure. To clarify this issue, that might not be clear in our first version of the manuscript, we now underline in the Methods section that all simulations started from a single configuration. About the possibility to compare the energy of several configurations relative to each other, we believe that this is not a recommended approach for molecular dynamics simulations of large molecules in explicit solvent. The reasons are: the presence of several minima, the difficulties to reach a well-converged optimization and the limitation of the force-field.

3-8) (5) RMSD figures of the MD-simulations can be provided to check the convergence of the MD simulations.

Following the suggestion of the referee, we added a plot of the Root Mean Square Displacement as a function of the simulated time as Supplementary Fig. 17.

(6) Units are used through each other (kcal/mol, meV).

We have unified the units. All energy values are provided in kcal/mol, throughout the paper.

3-9) (7) Can an agreement been found between the IR-spectra (Ext. Data Fig. 4), and calculated IR-spectra based on the studied model systems?

Unfortunately, this cannot be realistically targeted with the present computational capabilities. The comparison between experimental and theoretical IR-spectra along biocatalytic process can be achieved when the vibrational bands are well-separated and/or the 'signaling moieties' are clearly identified based on independent information. In our case, the likely source of vibrational bands are six carboxylate residues coordinated to metal ions of the Mn₄Ca-oxo cluster plus several further non-coordinated carboxylate residues of the D1/D2/CP43 chains within an estimated radius for relevant residues of 15 Å around the Mn₄Ca-oxo cluster. In this situation it is not possible to achieve a unique assignment of vibrational bands simply by comparing calculated and experimentally determined vibrational frequencies. Furthermore, the size of the relevant quantum region in conjunction with non-harmonic motions in H-bond clusters constitutes a major methodical problem. Spectra could be calculated directly by dipole-dipole autocorrelation function on QM/MM trajectories, but it would be necessary to simulate trajectories of lengths not accessible by the current computational facilities to resolve the spectra of an extended portion of the investigated system (at least D1/D2/CP43 chains) with the accuracy necessary (few cm⁻¹) to compare with the experimental data. A direct comparison is therefore, unfortunately, not possible.

3-10) There is appropriate use of statistics and treatment of uncertainties. For Ext. Data Fig. 5e; at some temperatures, there seem to be higher standard deviations on some of the measured time constants. The standard deviation of the time constants varies a lot for different temperatures. Is there any explanation for these variations?

The standard deviation itself is associated with an uncertainty (which might be described as the standard deviation of the standard deviation). We believe that the temperature-dependent differences do not reflect a trend but are explainable simply by statistical variations. Information on calculation of the standard deviations and the likely error in the experimentally determined energy figures is now provided in the caption of Extended Data Fig. 5.

3-11) E. Conclusions: robustness, validity, reliability

Conclusions are sound and suitable.

We are very pleased reading that the reviewer evaluates our conclusions as sound and suitable.

3-12) F. Suggested improvements: experiments, data for possible revision

Additional simulations and explanation of the computational part of the work (see under point C). Typos should be corrected in main text and supporting information.

A significant amount of additional simulations and explanations is provided in form of Supplementary Information, as detailed further above; typos have been corrected in main text, Extended Data Figure captions, and Supplementary Information.

3-13) G. References: appropriate credit to previous work?

The references appropriately credit previous work.

By a limited number of additional citations, we hope to have improved further in our coverage (and appreciation) of previous work.

3-14) H. Clarity and context: lucidity of abstract/summary, appropriateness of abstract, introduction and conclusions

The clarity and context are sufficient. However, some typos should be corrected, and certain sentences can be rewritten for extra clarity.

Typos have been corrected in main text, Extended Data Figure captions, and Supplementary Information. Especially in the figure captions and Supplementary Information, many sentences have been re-written for improved clarity, as suggested by the reviewer.

Referee #4:

The manuscript presents a spectacular time-resolved IR spectroscopic study founded on a setup that allowed performing FTIR step-scan experiments on PSII with data collection spanning very long times. The focus is on the final steps that reset the cycle from the S3 to the S0 state while forming and evolving dioxygen. Analysis of the data resolves the kinetics of the transition(s); a major point is the identification of a specific time constant with an “obligatory” step of proton removal. The manuscript contains a solid computational part that supports and extends the experimental work toward plausible mechanistic details of the transformations that may occur leading up to O-O bond formation. The work is exceptional in all aspects and contains a wealth of information as well as careful and thoughtful discussion. It is a significant contribution to the field and I have no essential comments regarding either the experimental or the computational components of the study.

We feel rewarded reading that the reviewer is highly appreciative of our study.

4-1) My main problem with the present work is the attribution of the label “S4” to the state associated with the obligatory proton removal. There is a fundamental difference in electronic structure depending on whether the hole is localized on the Mn4Ox cluster or on the tyrosine (Yz) that interfaces the cluster with P680. A metalloradical state of the OEC, i.e. a state with an oxidized tyrosyl radical (Yz*), is profoundly distinct in terms of its electronic and spectroscopic properties from a state with a reduced closed-shell Yz. These intermediates are clearly distinguished in approaches that are actually sensitive to where the electrons are located. For quite some time the S-state nomenclature is being used in (a large part of) the literature in a sense that reflects the electronic structure as understood in the above context (and not, for example, purely in the sense of kinetics). Thus, S0 and S0Yz* and S1 and S1Yz* and S2 and S2Yz* , and S3 and S3Yz* would all be distinct entities, both in theory and in practice (different properties, different observables – “pure” S-state versus “split signals” in EPR, X-ray spectroscopy etc). SiYz* versus S(i+1) simply means that in S(i+1) the hole has shifted from the tyrosyl radical to the Mn4Ox cluster – the OEC has transitioned. However, what the manuscript labels as “S4” is merely a form of S3Yz*. The S3Yz* intermediate is a classical metalloradical intermediate. This has been likely observed already by magnetic resonance spectroscopy, see e.g. FEBS Lett. 2014, 588:1827, and Photosynth. Res. 2016, 130:417. Is the EPR-observed intermediate the same as the here reported intermediate? This is an important question in itself that the authors must consider and comment upon. But whether this is the same or not, it is hard to justify calling it “S4” when the hole is still on Yz*. This would mean applying different nomenclature and different criteria for different use cases or for different experimental techniques. If we apply the same logic and the same criteria here as in all other transitions, then the only species discussed in this work that is really “S4” is the intermediate where an oxyl radical is formed on the cluster. This is also the same way such intermediates are discussed in the totality of the theoretical literature I am aware of. A deprotonation or any other event can of course occur while the system is poised electronically in a metalloradical Yz* state. This is well known for S2Yz*, where a lot of things do happen and several variants, including deprotonated forms, have been observed and characterized (see for example Boussac’s work on these intermediates). None of the various types of S2Yz* intermediates, deprotonated or not, would ever be called “S3”. Thus, using “S4” for a subsequent S3Yz* metalloradical intermediate, regardless of the protonation events and the kinetics involved, feels wrong to me.

We thank the reviewer for this insightfully elaborated and convincing arguments on the attribution of the label “S4”. In two invited and a third contributed talks, we presented the results of

our work at the International Photosynthesis Congress in New Zealand and its Oxygen Evolution satellite meeting (July/August 2022). The question how to name the intermediate states in the $S_3 \rightarrow S_0$ transition was also publicly discussed among the attending experts. *Following the consensus view expressed there and fully in line with the reviewer's suggestion, we have changed the nomenclature in Figure 8 and consequently throughout the manuscript (S_4 to S_3' , S_4' to S_4), as now also explained in the main text of the manuscript on page 6.* (The valuable articles on detection of likely closely related metallo-radical intermediates in low-temperature EPR experiments are now cited on pg. 2.)

Secondary points:

4-2) - Use chain identifiers when aminoacids are referred to, at least when they are mentioned for the first time. For example, is the Glu65–Glu212 pair in the D1 protein?

We now refer explicitly in the caption of Figure 1, to an extended version of Figure 1c presented in the Supplementary Information, where the chain identifiers as well as important atom labels of the Mn_4Ca -oxo cluster are provided.

4-3) - Page 3 mentions the Glu65-Glu212 pair, but Fig. 3 shows Glu65-Glu312.

With apologies, the erroneous use of Glu212 instead of Glu312 has been corrected in main article text and Supplementary Information.

4-4) - On p. 5 the phrase “Noteworthy, Yano and coworkers...” is accompanied by Ref. 21, but this is not the correct reference.

We now cite the correct references (Kern et al. 2018, Nature; Hussein et al. 2021, Nat. Commun. 2021) and apologize for the error.

4-5) - Page 5, “... which is in line with a mechanistic proposal [...] and recent structural data.”: Here Refs. 22 and 37 are cited, but the statement is not really supported by the references. Kern et al. is not a study that can support any correspondence between the published models and the present mechanism, while Ref. 37 is in fact inconsistent with the mechanism and the nature of intermediates proposed here. Suga et al. report a structural model (and accompanying calculations) that assign the presence of an oxyl radical in the S_3 state (i.e. they conclude that there is no Mn-centered oxidation in the S_2 - S_3 transition). This is not at all what the authors of the present manuscript propose.

- Also on p. 5, “We emphasize that Siegbahn and others...”: Here Ref. 40 is obviously a mistake; it is a study of pK values that is irrelevant here. It can be replaced by one of many studies on the mechanism by Siegbahn or others. I am not sure that the statement itself is entirely correct; one should have a closer look into some of the papers by Soji et al. (e.g. some 2018-2019 Chem. Phys. Letters).

We thank the referee very much for pointing out these inconsistencies. We have modified the references and the text on page 5/6 accordingly (plus clarifications in response to suggestions by the other reviewers).

Reviewer Reports on the First Revision:

Referees' comments:

Referee #2 (Remarks to the Author):

The manuscript has been edited to reflect my concerns as well as those of the other reviewers. I think it clearly lays out the conclusions which are well supported.

It's a pity that so much of the information needs to be in the SI, but this is necessary in an environment that rewards shorter main papers. The SI is quite substantial.

I do not have any additional comments.

Referee #3 (Remarks to the Author):

The manuscript has been extensively improved and is suitable for publication in Nature.

Referee #4 (Remarks to the Author):

All concerns and issues that were raised on the already excellent original submission have been fully addressed in the revised version.